# Breaking the Computational Barrier: Provably Efficient Actor–Critic for Low-Rank MDPs

**Ruiquan Huang** [1]  **Donghao Li** [2]  **Yingbin Liang** [3]  **Jing Yang** [2]

## Abstract

Reinforcement learning (RL) is a fundamental framework for sequential decision-making, in which an agent learns an optimal policy through interactions with an unknown environment. In settings with function approximation, many existing RL algorithms achieve favorable sample complexity, but often rely on computationally intractable oracles. In this paper, we use supervised learning as a computational proxy to establish a clear hierarchy of commonly adopted RL oracles under low-rank Markov Decision Processes (MDPs). This hierarchy shows that policy evaluation is the most computationally efficient oracle, provided that supervised learning can be efficiently solved. Motivated by this observation, we propose a novel optimistic actor-critic algorithm that relies solely on the policy evaluation oracle. We prove that our algorithm outperforms the existing sample complexity guarantees for low-rank MDPs while avoiding computationally expensive planning or optimization oracles commonly assumed in prior works. We further extend our theoretical results to approximately low-rank MDPs and demonstrate that this setting captures a broad class of real-world environments. Finally, we validate our theoretical results with experiments on several standard Gym environments.

## 1. Introduction

In reinforcement learning (RL), an agent interacts sequentially with an environment to optimize its decision-making strategy. A central challenge in RL is sample efficiency, which measures the number of interactions required to identify an optimal or near-optimal policy. Understanding sample complexity has been a core objective of RL theory, with extensive progress made under a variety of settings, including tabular MDPs with finite state–action spaces (Agrawal & Jia, 2017; Domingues et al., 2020; Ménard et al., 2021b), Linear MDPs (Jin et al., 2020; Wang et al., 2020), and more general function approximation frameworks (Jiang et al., 2017; Jin et al., 2021; Liu et al., 2022), where the state space may be infinite and the transition dynamics belong to rich function classes.

As deep learning continues to advance and apply to more complex real-world environments, RL with function approximation has become essential. However, in these settings, *sample efficiency* alone is no longer sufficient to characterize the practicality of an algorithm. *Computational efficiency* becomes a critical concern, since naively adopting sample-efficient algorithms under rich function classes often leads to procedures that are computationally intractable. As a result, many existing works rely on strong oracle assumptions, such as access to policy planning oracles (Uehara et al., 2021; Cheng et al., 2023; Agarwal et al., 2020) or constrained planning oracles (Modi et al., 2021; Mhammedi et al., 2023; Tan et al., 2025) that can be prohibitively expensive or even NP-hard in general function approximation settings. For instance, a constrained planning oracle requires computing the policy or action that maximizes a Q-function induced by a model class or confidence set. This gap between statistical (i.e., sample-complexity) guarantees and computational feasibility highlights the need for new algorithmic designs that are efficient in both senses.

In this paper, we focus on *low-rank Markov Decision Processes (MDPs)* (Agarwal et al., 2020), a structured yet expressive class of environments that has received significant attention in recent years. Low-rank MDPs assume that the transition kernel admits a low-rank factorization, meaning that the transition density can be represented using finite-dimensional embeddings of the state–action pair and the next state. This structural assumption captures a broad range of practical environments while enabling refined algorithmic and theoretical analysis. Our goal is to design an RL algorithm for low-rank MDPs that is *provably efficient* in both sample complexity and computational complexity.

A key challenge in formalizing computational efficiency

[1]University of Kentucky. [2]University of Virginia. [3]Ohio State University.. Correspondence to: Ruiquan Huang <rhu285@uky.edu>.

*Proceedings of the $43^{rd}$ International Conference on Machine Learning*, Seoul, South Korea. PMLR 306, 2026. Copyright 2026 by the author(s).

in RL theory is that traditional complexity measures, such as counting arithmetic operations, are often inadequate in function approximation settings. Instead, we adopt a widely used and practically meaningful proxy: *oracle complexity*. In particular, we take *standard supervised learning (regression)* as our basic computational primitive. Supervised learning represents one of the weakest and most commonly accepted oracle assumptions in modern machine learning. Our objective is therefore to design an RL algorithm whose computation can be entirely reduced to supervised learning subroutines. To justify this choice, we analyze the computational structure of commonly used RL oracles and establish a clear hierarchy among them, showing that planning-based oracles require solving strictly more complex optimization problems than policy evaluation, which in turn admits an efficient reduction to regression under low-rank MDPs.

This leads to the central question of this paper:

*Can we design a provably sample-efficient and computationally efficient algorithm for low-rank MDPs that relies only on supervised learning oracles?*

We provide an affirmative answer to the above question and make the following contributions.

- **Oracle Complexity.** We study three commonly adopted oracles in the RL literature: *policy evaluation*, *policy planning*, and *constrained planning*. We characterize their objective functions under low-rank MDPs and analyze key regularity properties, including convexity and smoothness. By establishing reduction relationships among these oracles, we demonstrate a strict hierarchy in oracle complexity, showing that planning-based oracles are computationally more demanding than policy evaluation, which can be efficiently implemented via supervised learning.

- **Algorithm.** We propose a new *optimistic actor–critic* (Opt-AC) algorithm for low-rank MDPs that relies solely on a policy evaluation oracle and therefore can be implemented entirely using supervised learning oracle. Our algorithm constructs an optimistic estimate of the Q-function for the current policy and updates the policy via mirror descent, avoiding explicit planning or constrained optimization steps.

- **Sample Complexity.** We prove that Opt-AC achieves a sample complexity of $\tilde{O}\left(\frac{H^5 d^3 |\mathcal{A}|^2 \log |\Theta|}{\epsilon^2}\right)$ for finding an $\epsilon$-optimal policy, where $H$ is the episode horizon, $d$ is the feature dimension, $|\mathcal{A}|$ is the number of actions, and $\Theta$ is the parameter space. This improves upon the best known result by a factor of $d$. The improvement is enabled by a refined error analysis that directly controls the empirical Gram matrix rather than relying on bounds derived from its expectation.

- **Bridging Theory and Practice.** We further show that a class of practical RL environments admit *approximately low-rank* transition structures. We extend our theoretical guarantees to approximate low-rank MDPs and implement Opt-AC in several standard Gym environments. Experimental results demonstrate that our algorithm consistently outperforms existing baselines across most tasks.

## 2. Related Work

There is a long line of work on the theoretical foundations of RL (Auer et al., 2008; Azar et al., 2017; Jin et al., 2018; Jiang et al., 2017; Bai et al., 2019; Jin et al., 2020; Ménard et al., 2021a; Uehara et al., 2021; Cheng et al., 2023; Zhang et al., 2022b). Many existing algorithms achieve favorable sample complexity and computational efficiency simultaneously in structured settings such as *tabular* and *linear* MDPs. In these regimes, computational complexity is typically measured by the number of arithmetic operations. For example, algorithms for tabular MDPs often incur computational complexity on the order of $O(H|\mathcal{S}||\mathcal{A}|K)$ (Ménard et al., 2021a), while algorithms for linear MDPs typically scale as $O(Hd|\mathcal{A}|K)$ (Jin et al., 2020). In particular, Lin et al. (2025) designed practical and computationally efficient algorithms through randomized gradient descent for linear MDPs. However, when extending beyond linear structure to *low-rank MDPs* and more general function approximation settings, existing works often fail to explicitly characterize computational complexity, or rely on algorithmic primitives that are difficult to implement in practice.

**Low-Rank MDPs.** Low-rank MDPs were first studied by Agarwal et al. (2020), who proposed a reward-free algorithm aimed at learning transition dynamics that enable planning under arbitrary reward functions. Subsequently, Uehara et al. (2022) proposed the REP-UCB algorithm for reward-known low-rank MDPs and achieved a sample complexity of $\tilde{O}\left(\frac{H^7 d^4 |\mathcal{A}|^2 \log |\Theta|}{\varepsilon^2}\right)$. Later, Cheng et al. (2023) improved this bound to $\tilde{O}\left(\frac{H^5 d^4 |\mathcal{A}|^2 \log |\Theta|}{\varepsilon^2}\right)$ in the reward-free setting by introducing a clipping technique. Zhang et al. (2021) proposed ReLEX for a closely related low-rank model. Despite these improved sample complexity guarantees, the above approaches rely heavily on *policy planning oracles*, which require solving difficult optimization problems and are usually training-unstable in practice. Several subsequent works (Ren et al., 2022; Zhang et al., 2022a) attempted to bridge theory and practice by exploiting the representational structure of low-rank MDPs and redesigning the representation learning component. Nevertheless, these methods still depend on policy planning oracles, leading to a mismatch between theoretical assumptions and practical implementations. Model-free algorithms have also been proposed for low-rank MDPs (Modi et al., 2021; Mhammedi et al.,

| Name | Oracle Types | Convexity | Smoothness | Computation | Sample Complexity |
|---|---|---|---|---|---|
| **FLAMBE** (Agarwal et al., 2020) | PP($\varepsilon$) | ✗ | ✗ | $H\mathsf{SL}(\varepsilon^2\|\mathcal{A}\|^{-2H})$ | $\tilde{O}\left(\frac{H^{22}d^7\|\mathcal{A}\|^9\log\|\Theta\|}{\epsilon^{10}}\right)$ |
| **Rep-UCB** (Uehara et al., 2021) | PP($\varepsilon$) | ✗ | ✗ | $H\mathsf{SL}(\varepsilon^2\|\mathcal{A}\|^{-2H})$ | $\tilde{O}\left(\frac{H^7d^4\|\mathcal{A}\|^2\log\|\Theta\|}{\epsilon^2}\right)$ |
| **RAFFLE** (Cheng et al., 2023) | PP($\varepsilon$) | ✗ | ✗ | $H\mathsf{SL}(\varepsilon^2\|\mathcal{A}\|^{-2H})$ | $\tilde{O}\left(\frac{H^5d^4\|\mathcal{A}\|^2\log\|\Theta\|}{\epsilon^2}\right)$ |
| **SpanRL** (Mhammedi et al., 2023) | CP($\varepsilon$) | ✗ | ✗ | $2^d\mathsf{SL}\left(\varepsilon^2\|\mathcal{A}\|^{-2H}\right).$ | $\tilde{O}\left(\frac{H^6d^9\|\mathcal{A}\|^4\log\|\Phi\|}{\epsilon^2}\right)$ |
| **MOFFLE** (Modi et al., 2021) | CP($\varepsilon$) | ✗ | ✗ | $2^d\mathsf{SL}\left(\varepsilon^2\|\mathcal{A}\|^{-2H}\right).$ | $\tilde{O}\left(\frac{H^7d^{11}\|\mathcal{A}\|^{14}\log\|\Phi\|}{\epsilon^2}\right)$ |
| **Opt-AC** (Ours) | PE($\varepsilon$) | ✓ | ✓ | $\mathsf{SL}(\varepsilon^2)$ | $\tilde{O}\left(\frac{H^5d^3\|\mathcal{A}\|^2\log\|\Theta\|}{\epsilon^2}\right)$ |
| **Lower Bound** (Cheng et al., 2023) | - | - | - | - | $\Omega\left(\frac{Hd\|\mathcal{A}\|}{\epsilon^2}\right)$ |

*Table 1.* Comparison of algorithms for low-rank MDPs in terms of oracle structure, computational properties, and sample complexity. *Oracle Types* indicate the highest-level optimization oracle required (policy evaluation PE, policy planning PP, or constrained planning CP). *Convexity* and *Smoothness* refer to the regularity of the objective optimized by the corresponding oracle. *Computation* reports the best-known $\mathsf{SL}$-oracle complexity required to implement the oracle, expressed using the summary $\mathrm{SL}^m(\varepsilon)$ (Definition 3.6). For clarity, coverage constants are omitted. All results are stated for low-rank MDPs with unknown features. Here $d$ is the feature dimension, $\|\mathcal{A}\|$ is the action space size, and $H$ is the horizon. $\Theta = \Phi \times \Psi$ is the parameter space. The accuracy parameter $\varepsilon$ in the computation column corresponds (up to constants and logarithmic factors) to the inverse square root of the sample complexity.

2023), but these methods rely on even stronger *constrained planning oracles*. From a computational perspective, Rohatgi & Foster (2025) analyze the necessary oracle (also use regression as a proxy) for block MDPs, a special case of low-rank MDPs, and identify fundamental computational hardness results. Their findings complement our work by presenting a computational limits of low-rank MDPs.

**General Function Approximation.** Beyond low-rank MDPs, a variety of structural assumptions have been proposed to enable sample-efficient RL with general function approximation, including low Bellman or witness rank (Jiang et al., 2017; Sun et al., 2019), bilinear function classes (Du et al., 2021), and low Bellman eluder dimension (Jin et al., 2021). These frameworks subsume low-rank MDPs as special cases and often yield sharp sample complexity guarantees. However, algorithms in these settings are typically computationally intractable (Uehara et al., 2022). Another closely related work is Tan et al. (2025), which developed an actor–critic algorithm for general function approximation with low Bellman eluder dimension and achieved a sample complexity of $\tilde{O}\left(\frac{dH^5+dH^4\log\|\Theta\|}{\varepsilon^2}\right)$. However, this approach also relies on the strongest form of optimization oracle, namely constrained planning.

## 3. Problem Setting

**Notation.** For any integer $H \in \mathbb{N}$, let $[H] := \{1,\ldots,H\}$. For a vector $x \in \mathbb{R}^d$, $\|x\|_2$ denotes the Euclidean norm. For a positive semidefinite matrix $A \in \mathbb{R}^{d\times d}$, we define $\|x\|_A := \sqrt{x^\top A x}$. For two probability measures $P$ and $Q$, $\mathsf{D}_{\mathsf{KL}}(P\|Q)$ denotes their KL-divergence. Throughout, we use $\widetilde{O}(\cdot)$ to hide logarithmic factors.

### 3.1. RL Formulation

**Episodic Markov Decision Processes.** We consider a finite-horizon episodic Markov decision process (MDP) $\mathcal{M} = (\mathcal{S}, \mathcal{A}, H, \{\mathcal{T}_{\theta,h}\}_{h=1}^H, \{r_h\}_{h=1}^H)$, where $\mathcal{S}$ is a (possibly infinite) state space, $\mathcal{A}$ is a finite action space, $H \in \mathbb{N}$ is the episode length, $\mathcal{T}_{\theta,h}(\cdot \mid s,a)$ is a time-dependent transition kernel parameterized by $\theta \in \Theta$, and $r_h : \mathcal{S}\times\mathcal{A} \to [0,1]$ is a deterministic known reward function. Each episode starts from a fixed initial state $s_1$.

A (non-stationary) policy $\pi = \{\pi_h\}_{h=1}^H$ consists of mappings $\pi_h : \mathcal{S} \to \Delta(\mathcal{A})$. At step $h$, the agent observes $s_h$, selects $a_h \sim \pi_h(\cdot \mid s_h)$, receives reward $r_h(s_h, a_h)$, and transitions to $s_{h+1} \sim \mathcal{T}_{\theta,h}(\cdot \mid s_h, a_h)$.

For a policy $\pi$ and model parameter $\theta$, the value and action-value functions are defined as

$$V_{\theta,r,h}^\pi(s) := \mathbb{E}_{\pi,\theta}\left[\sum_{t=h}^H r_t(s_t, a_t) \,\middle|\, s_h = s\right],$$
$$Q_{\theta,r,h}^\pi(s,a) := r_h(s,a) + \mathbb{E}_{s'\sim\mathcal{T}_{\theta,h}(\cdot\mid s,a)}\left[V_{h+1,\theta}^\pi(s')\right],$$

with terminal condition $V_{\theta,r,H+1}^\pi(\cdot) \equiv 0$. We write $V_\theta^\pi := V_{\theta,r,1}^\pi(s_1)$. Let $\theta^\star$ denote the true model parameter, and let $V^\star := \sup_\pi V_{\theta^\star}^\pi$ be the optimal value. Our goal is to learn an $\epsilon$-optimal policy $\bar{\pi}$ such that $V^* - V_{\theta^*,r}^{\bar{\pi}} \leq \epsilon$.

Throughout, expectations $\mathbb{E}_\theta^\pi[\cdot]$ are taken with respect to the trajectory distribution induced by policy $\pi$ and transition model $\mathcal{T}_\theta$, and $Q_{\theta,r,h}^*(s,a)$ is the optimal Q-value function.

**Low-Rank MDPs.** We study episodic MDPs whose transition dynamics admit an exact low-rank factorization with unknown feature representations. For each step $h \in [H]$, the transition kernel $\mathcal{T}_{\theta,h}$ is said to be *low-rank with dimension* $d$ if there exist latent feature maps $\phi_{\theta,h} : \mathcal{S} \times \mathcal{A} \to \mathbb{R}^d$ and $\mu_{\theta,h} : \mathcal{S} \to \mathbb{R}^d$ such that $\forall (s,a,s') \in \mathcal{S} \times \mathcal{A} \times \mathcal{S}$, we have

$$\mathcal{T}_{\theta,h}(s' \mid s,a) = \langle \phi_{\theta,h}(s,a), \mu_{\theta,h}(s') \rangle.$$

We assume the normalization conditions $\|\phi_{\theta,h}(s,a)\|_2 \leq 1$ and $\left\| \int \mu_{\theta,h}(s') g(s') \, ds' \right\|_2 \leq \sqrt{d}$ for all measurable $g : \mathcal{S} \to [0,1]$. An MDP is called a low-rank MDP if this factorization holds for all $h \in [H]$.

The feature representations $\{\phi_{\theta,h}, \mu_{\theta,h}\}$ are latent and unknown to the agent, so learning the transition model requires learning these representations from data. We adopt a realizability assumption: the true transition and reward models belong to known function classes, which is standard in low-rank and general function approximation settings (Jin et al., 2020; Uehara et al., 2021; Jin et al., 2021).

**Assumption 3.1** (Realizability). The agent is given function classes $\Phi$ and $\Psi$ such that $\phi_{\theta^\star,h} \in \Phi, \mu_{\theta^\star,h} \in \Psi, \forall h \in [H]$.

For simplicity of exposition, we assume the model class $\Theta := \Phi \times \Psi$ is finite. This is used to simplify confidence bounds and concentration arguments; extensions to infinite classes with bounded statistical complexity follow standard covering number techniques and are omitted.

### 3.2. Optimization Oracles and Computational Model

To formalize computational efficiency, we measure complexity through access to *optimization oracles*, a standard abstraction in theoretical reinforcement learning. Each oracle is specified by the functional objective it must solve and the accuracy criterion under which success is measured. We focus on four oracles commonly adopted in the RL literature and characterize their complexity solely through SL-oracle complexity.

**Definition 3.2** (Supervised Learning Oracle). A supervised learning oracle $\mathsf{SL}(\varepsilon)$ takes as input a parameter space $\mathcal{W}$, a data distribution (sampling oracle) $\rho$ over input-output pairs $(x,y)$, and a loss function $\ell : \mathcal{Y}^2 \to \mathbb{R}_+$. It returns a parameter $\hat{w} \in \mathcal{W}$ such that

$$\mathbb{E}_{(x,y)\sim\rho}[\ell(f_{\hat{w}}(x), y)] \leq \inf_{w \in \mathcal{W}} \mathbb{E}_{(x,y)\sim\rho}[\ell(f_w(x), y)] + \varepsilon.$$

We emphasize that $\mathsf{SL}(\varepsilon)$ models a single-level risk minimization problem with a fixed hypothesis class $\mathcal{W}$ and a simple loss function $\ell$ (e.g. L2 or cross-entropy loss); in particular, it does not include constrained or bilevel optimization where the hypothesis class or loss depends on data, input or another optimization problem.

**Definition 3.3** (Policy Evaluation Oracle). A policy evaluation oracle $\mathsf{PE}(\varepsilon)$ takes as input a model $\theta$, a policy $\pi$, and a reward function $r$, and returns an action-value function $\{\hat{Q}_h\}_{h=1}^H$ such that, for a specified distribution $\rho$,

$$\mathbb{E}_{(s,a)\sim\rho}\left[ \left| Q_{\theta,r,h}^\pi(s,a) - \hat{Q}_h(s,a) \right| \right] \leq \varepsilon \quad \text{for all } h.$$

**Definition 3.4** (Policy Planning Oracle). A policy planning oracle $\mathsf{PP}(\varepsilon)$ takes as input a model $\theta$ and reward function $r$, and returns an action-value function $\{\hat{Q}_h\}_{h=1}^H$ satisfying, for a specified distribution $\rho$,

$$\mathbb{E}_{(s,a)\sim\rho}\left[ \left| Q_{\theta,r,h}^\star(s,a) - \hat{Q}_h(s,a) \right| \right] \leq \varepsilon \quad \text{for all } h.$$

**Definition 3.5** (Constrained Planning Oracle). A constrained planning oracle $\mathsf{CP}(\varepsilon)$ is given a reward function $r$ and a constraint set over models $\Theta_c := \{\theta : L(\theta) \leq c\}$, where $L(\theta)$ is a supervised learning loss. It returns an action-value function $\hat{Q}$ such that, for a specified distribution $\rho$,

$$\mathbb{E}_{(s,a)\sim\rho}\left[ \left| \hat{Q}_h(s,a) - \sup_{\theta \in \Theta_c} Q_{\theta,h}^\star(s,a) \right| \right] \leq \varepsilon \quad \text{for all } h.$$

**SL-oracle Complexity.** We measure computational efficiency through the number and accuracy of calls to supervised learning oracles, treating supervised learning as the basic computational primitive. This abstraction allows us to compare different reinforcement-learning primitives without committing to a specific optimization algorithm or implementation.

**Definition 3.6** (SL-oracle complexity). Consider an algorithm that invokes supervised learning oracles $\mathsf{SL}(\varepsilon_1), \ldots, \mathsf{SL}(\varepsilon_m)$. We define its SL-oracle complexity by $\mathrm{SL}^m(\varepsilon_{\min})$, where $\varepsilon_{\min} := \min_{i \in [m]} \varepsilon_i$. This definition records the total number of oracle calls and the most stringent accuracy requirement.

With Definition 3.6, we present the following proposition that builds a hierarchy among $\mathsf{PE}, \mathsf{PP}$, and $\mathsf{CP}$ in terms of SL-oracle-complexity. The proof can be found in Appendix A.

**Proposition 3.7** (Best-known SL-oracle complexity). *Under low-rank MDP settings, suppose the data distribution $\rho$ has coverage constant $C$ with respect to the policy $\pi$ considered in $\mathsf{PE}$ and uniform policy $\mathrm{u}$, i.e.,*

$$\max_{\pi' \in \{\pi, \mathrm{u}\}} \mathbb{E}_{(s,a)\sim\rho}\left[ \mathcal{T}_h(s' \mid s,a)\pi'(a' \mid s') \right] \leq C \, \rho(s',a').$$

*Then the following best-known SL-oracle complexity bounds hold for achieving an $\varepsilon$-approximate solution.*

*1. $\mathsf{PE}(\varepsilon) : \mathsf{SL}^1\left( \tilde{O}(\varepsilon^2 C^{-2H}) \right).$*

*2. $\mathsf{PP}(\varepsilon) : \mathsf{SL}^H\left( \tilde{O}(\varepsilon^2 (|\mathcal{A}|C)^{-2H}) \right).$*

*3.* $\mathsf{CP}(\varepsilon) : \mathsf{SL}^{\Omega(2^d)}\left(\tilde{O}\left(\varepsilon^2(|\mathcal{A}|C)^{-2H}\right)\right).$

*Moreover, only policy evaluation's loss function is convex and smooth.*

Intuitively, policy evaluation corresponds to a convex and smooth optimization problem, while policy planning is non-smooth due to the Bellman optimality operator and requires a backward solver with $H$ stages. Constrained planning further induces a bilevel optimization structure, for which only stationary-point guarantees are known (Huang et al., 2022; Beck et al., 2023), and global optimization is believed to require exponentially many supervised learning oracle calls[1].

# 4. Algorithm: Optimistic Actor–Critic with Representation Learning

In this section, we present an optimistic actor–critic algorithm for low-rank MDPs. The algorithm proceeds in episodes. At each iteration, it (i) collects exploratory data using the current policy, (ii) learns a transition model via maximum likelihood estimation, (iii) constructs an optimistic critic by incorporating an exploration bonus, and (iv) updates the policy through an actor step.

Before introducing the components of the algorithm in detail, we summarize the hyperparameters and oracle access required. Let $K$ denote the total number of iterations and let $\epsilon > 0$ denote the target accuracy, corresponding to an $\epsilon$-optimal policy. Our algorithm relies on two oracles: (i) a policy evaluation oracle, which can be implemented via supervised learning, and (ii) a sampling oracle $\rho$ over the state-action space. The sampling oracle $\rho$ is required to provide sufficient coverage of the optimal policy's occupancy measure. Specifically, letting $d^{\star}_{\theta,h}(s)$ denote the marginal state distribution induced by the enviroment $\theta$ and the optimal policy $\pi^{\star}$ at step $h$, we assume that there exists a finite constant $C$ such that $\max\{d^{\star}_{\theta^*,h}(s), d^{\star}_{\hat{\theta}_k,h}(s)\}\mathtt{u}(a) \leq C\,\rho(s,a), \forall(s,a,h)$, where $\mathtt{u}$ denotes the uniform distribution over actions, $\hat{\theta}_k$ is the estimated model at iteration $k$ (defined later). This assumption is mild and can be satisfied, for example, by choosing $\rho$ as a uniform distribution over the state-action space when the state space is compact (Towers et al., 2024).

**Difference to Existing Algorithms.** Our design differs fundamentally from existing approaches for low-rank MDPs in both algorithmic structure and oracle requirements. **(a) Low-computation-cost Oracle.** In contrast to prior meth-

---

[1]Proposition 3.7 summarizes a hierarchy of *best-known oracle upper bounds and optimization guarantees*. It does not preclude the existence of more efficient reductions for PP/CP, and establishing oracle lower bounds is an open problem.

ods that rely on policy planning or constrained planning oracles, our algorithm requires only a *policy evaluation* oracle, which can be efficiently implemented via supervised learning due to Proposition 3.7. **(b) Approximate Oracle Solver.** Moreover, we do not assume access to exact oracle solutions. Instead, our analysis accommodates a more realistic *sample-then-optimize* setting, in which all optimization steps can utilize finite data and approximate solvers. **Actor-critic Style Design.** Our method departs from pure UCB-style designs and adopts an optimistic actor-critic framework. The optimistic critic is constructed directly from the learned model without invoking any global planning subroutine, and the actor update admits an exact-implementable method. As a result, every component of the algorithm is implementable using standard supervised learning and sampling oracles. The pseudocode is presented in Algorithm 1.

**Collect Exploratory Data.** Suppose we are at iteration $k$ and have a policy $\pi^{(k)}$. For each $h$, execute policy $\pi^{(k)}$ up to step $h-1$, then execute uniform policy $\mathtt{u}$ for step $h-1, h$, and obtain trajectory: $\left(s^{k,h}_1, a^{k,h}_1, \ldots, s^{k,h}_{h-1}, a^{k,h}_{h-1}, s^{k,h}_h, a^{k,h}_h, s^{k,h}_{h+1}\right)$. The idea is that for episode $(k,h)$, $a_{h-1}$ and $a_h$ are selected uniformly randomly. The second uniform exploration facilitates the transition estimation at step $h$ and thus the feature extraction, while the first uniform exploration helps to define the uncertainty at step $h-1$.

**Representation Learning via Maximum Likelihood.** To learn the latent transition structure, the agent maintains a dataset $\mathcal{D}^{(k)}_h$ consisting of observed transitions at step $h$ up to iteration $k$. Given the collected data, we estimate the model parameter $\theta \in \Theta$ by minimizing the negative log-likelihood:

$$\mathcal{L}^{(k)}(\theta) = -\sum_{n<k} \log \mathcal{T}_{\theta,h}(s^{n,h}_{h+1} \mid s^{n,h}_h, a^{n,h}_h). \quad (1)$$

Rather than requiring exact minimization, we allow approximate empirical risk minimization. Specifically, the estimator $\hat{\theta}_k$ satisfies $\mathcal{L}^{(k)}(\hat{\theta}_k) \leq \min_{\theta\in\Theta} \mathcal{L}^{(k)}(\theta) + \beta$, where $\beta$ is a confidence width that scales logarithmically with the model class size and the total number of samples. Note that this step only requires one supervised learning oracle $\mathsf{SL}$ call.

**Optimistic Critic via Exploration Bonus.** To encourage efficient exploration, we augment the reward with an optimism-driven exploration bonus. Specifically, for each step $h$, we define the empirical Gram matrix

$$\hat{\Lambda}_{k,h} = \lambda I + \sum_{n<k} \hat{\phi}^k_h(s^{n,h+1}_h, a^{n,h+1}_h)\hat{\phi}^k_h(s^{n,h+1}_h, a^{n,h+1}_h)^{\top},$$

where $\hat{\phi}^k_h = \phi_{\hat{\theta}_k,h}$ is the learned feature of $\hat{\theta}_k$. The raw exploration bonus is defined as

$$\tilde{b}^{(k)}_h(s,a) = \min\left\{\alpha\|\phi_{\hat{\theta}_k,h}(s,a)\|_{\hat{\Lambda}^{-1}_{k,h}}, 1\right\}. \quad (2)$$

The term $\|\hat{\phi}_k(s,a)\|_{\hat{\Lambda}_{k,h}^{-1}}$ captures the statistical uncertainty of the learned representation in direction $(s,a)$, analogous to elliptical confidence bounds in linear bandits. We set the actual exploration bonus function as $\hat{b}_h^{(k)} = 3H\tilde{b}_h^{(k)}$. The multiplicative scaling is chosen so that the bonus term dominates the propagation of one-step transition estimation error through the $H$-step Bellman recursion. As a result, the optimistic critic constructed using $r + \hat{b}^{(k)}$ upper-bounds the true value function with high probability.

Using the learned model $\hat{\theta}_k$, the augmented reward $r + \hat{b}^{(k)}$, and the current policy $\pi^{(k)}$, the critic $Q_{\hat{\theta}_k, r+\hat{b}^{(k)}, h}^{\pi^{(k)}}(s,a)$ is estimated by calling the policy evaluation oracle $\mathsf{PE}(1/\sqrt{K})$. So we obtain $\{\hat{Q}_{k,h}\}_{h=1}^{H}$ that satisfies $\forall h$,

$$\mathbb{E}_{(s,a)\sim\rho}\left[\left|\hat{Q}_{k,h}(s,a) - Q_{\hat{\theta}_k, r+\hat{b}^{(k)}, h}^{\pi^{(k)}}(s,a)\right|\right] \leq \frac{1}{\sqrt{K}}. \quad (3)$$

**Actor Update via KL-Regularized Policy Improvement.**
Given the optimistic critic, the policy is updated via a KL-regularized actor step. For a reference policy $\pi^{(k)}$ and learning rate $\eta > 0$, we define the actor objective

$$\mathcal{L}_a^{(k)}(\pi) = \left\langle \pi_h(\cdot \mid s), \hat{Q}_{k,h}(s,\cdot) \right\rangle \\ - \frac{1}{\eta}\mathsf{D}_{\mathrm{KL}}\left(\pi_h(\cdot \mid s) \,\|\, \pi_h^{(k)}(\cdot \mid s)\right). \quad (4)$$

This objective is strictly concave in $\pi_h(\cdot \mid s)$ and admits a closed-form solution. While one could alternatively invoke a supervised learning oracle to approximately solve (4), for clarity of presentation and analysis we use its exact solution:

$$\pi_h^{(k+1)}(a \mid s) \propto \pi_h^{(k)}(a \mid s)\exp\left(\eta\,\hat{Q}_{k,h}(s,a)\right). \quad (5)$$

This update can be implemented efficiently in polynomial time. In particular, it suffices to store the estimated critics $\{\hat{Q}_{k,h}\}_{k,h}$, and at interaction time to sample $a_h$ according to a softmax distribution induced by $\eta\sum_{k'\leq k}\hat{Q}_{k',h}(s_h,\cdot)$. Importantly, the actor update does not require solving a global planning problem, which distinguishes our approach from prior UCB-style algorithms that rely on policy or constrained planning oracles.

## 5. Theoretical Results

In this section, we present our main results.

**Theorem 5.1** (Computational Complexity). *The $\mathsf{SL}$-oracle complexity of Algorithm 1 is at most $\mathsf{SL}^{2K}(1/K)$.*

*Proof.* We analyze the computational cost of the proposed algorithm and show that it remains lightweight. The main computational overhead arises in Steps 5–7 of Algorithm 1. Step 4 (model learning) is a standard supervised learning

---

**Algorithm 1** Optimistic Actor–Critic (Opt-AC)
1: Initialize policy $\pi^{(0)}$ to be uniform over $\mathcal{A}$. Set hyperparameters: maximum iteration number $K$, error rate $\epsilon$, coefficients $\beta, \alpha, \lambda$.
2: **for** $k = 0, 1, \ldots, K$ **do**
3:     Collect $H$ exploratory trajectories using $\pi^{(k)}$.
4:     Learn model $\hat{\theta}_k$ via approximate MLE. (SL)
5:     Construct exploration bonus $\hat{b}^{(k)}$ via (2).
6:     Estimate optimistic critic $Q_{\hat{\theta}_k, r+\hat{b}^{(k)}, h}^{\pi^{(k)}}$ by (3). (PE)
7:     Update policy $\pi^{(k+1)}$ by (5).
8: **end for**
9: Return policy $\bar{\pi} = \mathrm{Uniform}\{\pi^{(0)}, ..., \pi^{(k)}\}$

---

problem and its oracle complexity is $\mathsf{SL}(\beta)$. Step 5 constructs the empirical Gram matrices using the learned feature representations; this step requires only a single pass over the collected data and does not involve any iterative optimization, incurring only additional memory to store the matrices. Step 6 performs policy evaluation, whose oracle complexity is $\mathsf{SL}(1/K)$ due to Proposition 3.7. Since the total iteration of Algorithm 1 is $K$ and most stringent accuracy requirement is $1/K > \beta$, by Definition 3.6, the SL-oracle complexity is $\mathsf{SL}^{2K}(1/K)$. $\qquad\square$

Then, we state our main theorem for sample complexity. Proofs can be found in Appendix B.

**Theorem 5.2** (Sample Complexity). *Assume $\mathcal{M}$ is a low-rank MDP with dimension $d$, and Assumption 3.1 holds. Given any $\epsilon, \delta \in (0,1)$, let $\bar{\pi}$ be the output of Opt-AC and $\pi^\star$ be the optimal policy under the true model $P^\star$. Set $\beta = O(\log(K|\Theta|/\delta))$, $\alpha = \tilde{O}(\sqrt{|\mathcal{A}|})$, $\eta = O(\frac{1}{H\sqrt{K}})$ and $\lambda = \tilde{O}(1/d)$. Then, with probability at least $1 - \delta$, we have $V_{P^\star, r}^{\pi^\star} - V_{P^\star, r}^{\bar{\pi}} \leq \epsilon$, and the total number of trajectories collected by Opt-AC is upper bounded by $\tilde{O}(\frac{H^5 d^3 |\mathcal{A}|^2 \log|\Theta|}{\epsilon^2})$.*

Compared with RAFFLE (Cheng et al., 2023), our result improves their sample complexity of $H^3 d^4 |\mathcal{A}|^2$ by a factor of $d$. Note that $d^3$ is also a common order in linear MDP setting where the feature is known (Jin et al., 2020). The improvement is brought by a more careful analysis to the one-step back inequality. One-step back inequality is a standard technique in low-rank MDP literature. It shows that the optimistic $Q$-value, $Q_{\hat{\theta}_k, r+\hat{b}^{(k)}, h}^{\pi}$, is indeed an upper bound of the estimation error of $\hat{\theta}_k$. One of the key factors that affect the sample complexity is the coefficient of $\|\phi(s_h, a_h)\|_{\hat{\Lambda}_{k,h}^{-1}}$ in $\hat{b}^{(k)}$. Here our choice is $\tilde{O}(H\sqrt{|\mathcal{A}|})$, while existing works often choose $\tilde{O}(H\sqrt{d^2 + |\mathcal{A}|})$ (Uehara et al., 2021; Cheng et al., 2023), which leads to a higher-order dependence on $d$. Our analysis also differs fundamentally from that of Tan et al. (2025), who study ac-

tor–critic methods using a CP oracle over a general function class. In contrast, operating in the low-rank MDP setting and *explicit bonus* design requires substantially more delicate control of the evolution of the gap between the optimal value function and the optimistic critic value. This gap directly affects policy updates and can accumulate over time if not carefully bounded, making actor–critic-style analysis significantly more challenging in our setting. Despite these difficulties, our approach achieves the best known sample complexity guarantees for low-rank MDPs while relying only on supervised learning oracles.

## 6. Results for Misspecified Setting

An MDP is $\zeta$-approximate low-rank if there exists $\zeta > 0$ and $\theta \in \Theta$ such that for all $(s_h, a_h)$, we have

$$\left| \mathcal{T}_{\theta,h}(s_{h+1} \mid s_h, a_h) - \langle \phi_{\theta,h}(s_h, a_h), \mu_{\theta,h}(s_{h+1}) \rangle \right| \leq \zeta.$$

**Richness of Approximate Low-rank MDPs.** We provide a formal justification showing that the class of $\zeta$-approximate low-rank MDPs is expressive and encompasses a broad family of MDPs commonly encountered in practice.

*Gaussian transition kernels.* As a motivating example, consider Gaussian transition kernels $\mathcal{T}_{\theta,h}(\cdot \mid s, a) = \mathcal{N}(f_h(s, a), I)$. Such transitions admit approximate low-rank representations via random Fourier features (Rahimi & Recht, 2007), where the dependence on $(s, a)$ enters through the mean function $f_h(s, a)$ and the next-state $s'$ is encoded using a shared Fourier basis. Standard concentration results imply that the transition density can be approximated uniformly by an inner product $\langle \phi_h(s, a), \mu_h(s') \rangle$ with error $O(d^{-1/2})$ on compact domains. Consequently, choosing $d = O(\zeta^{-2})$ yields a $\zeta$-approximate low-rank representation, a construction that has been exploited in prior work such as Ren et al. (2022).

*Beyond Gaussian transitions.* While random Fourier features are classically associated with Gaussian kernels, the approximation framework extends to a much broader class of conditional distributions satisfying mild regularity conditions. In particular, we explicitly construct an approximation algorithm cRFF (Algorithm 2) and characterize its error guarantees in Theorem 6.1. This result complements our theoretical analysis and further bridges the gap between the low-rank MDP abstraction and practical transition models. The proof can be found in Appendix E.

**Theorem 6.1** (Approximate low-rank representation)**.** *Let $\mathcal{S} \subset \mathbb{R}^D$ be compact. For each $(\theta, h) \in \Theta \times [H]$ and $(s, a) \in \mathcal{S} \times \mathcal{A}$, assume access to a sampling oracle for $\mathcal{T}_{\theta,h}(\cdot \mid s, a)$ that provides $N$ i.i.d. next-state samples. Suppose the transition kernels satisfy the following conditions.*

*(A1) Boundedness. There exists an $M > 0$ such that $\sup_{\theta,s,a,s',h} \mathcal{T}_{\theta,h}(s' \mid s, a) \leq M$.*

---

**Algorithm 2** Conditional Random Fourier Features (cRFF)

1: **Input:** $B_W = \{w \in \mathbb{R}^D : \|w\| \leq W\}$.
2: Draw $d$ i.i.d. frequencies $w_1, \ldots, w_d$ uniformly from $B_W$. Denote the volume of $B_W$ by $\mathrm{Vol}(B_W)$.
3: Define the feature map $\mu : \mathcal{S} \to \mathbb{R}^{2d}$, where $[\mu(y)]_{2k} = \cos(2\pi w_k^\top s), [\mu(y)]_{2k+1} = -\sin(2\pi w_k^\top s), \forall k \in [d]$
4: Draw $N$ samples $y^{(1)}, \ldots, y^{(N)} \sim \mathcal{T}_{\theta,h}(\cdot \mid s, a)$. For each $k \in [d]$, compute the empirical Fourier coefficient $\hat{g}_k = \frac{1}{N} \sum_{n=1}^{N} e^{-2\pi i w_k^\top y^{(n)}}$, and let $\hat{g}_k^{\mathrm{r}}, \hat{g}_k^{\mathrm{i}}$ be the real and imaginary part of $\hat{g}_k$, respectively.
5: Return $\hat{\phi}_\theta(s, a) \in \mathbb{R}^{2d}$ as

$$\hat{\phi}_\theta(s, a) = \left( \hat{g}_1^{\mathrm{r}}, \hat{g}_1^{\mathrm{i}}, \ldots, \hat{g}_d^{\mathrm{r}}, \hat{g}_d^{\mathrm{i}} \right) \times \mathrm{Vol}(B_W)/\sqrt{d}.$$

---

*(A2) Smoothness and boundary vanishing conditions. There exist an integer $m > D$ and such that $\mathcal{T}_{\theta,h}(\cdot \mid s, a)$ is $m$-times continuously differentiable on $\mathcal{S}$, and all derivatives up to order $m - 1$ vanish on the boundary $\partial \mathcal{S}$.*

*Then, for any $\zeta > 0$, there exists a randomized algorithm cRFF (Algorithm 2) that constructs $2d(\zeta)$-dimensional feature maps $\hat{\phi}_\theta(s, a), \mu(s') \in \mathbb{R}^{2d}$ using $N(\zeta)$ samples such that, with probability at least $1 - \delta$, uniformly over all $(\theta, s, a, s', h) \in \Theta \times \mathcal{S}^2 \times \mathcal{A} \times [H]$,*

$$\left| \mathcal{T}_{\theta,h}(s' \mid s, a) - \hat{\phi}_\theta(s, a)^\top \mu(s') \right| \leq \zeta.$$

*Here $d(\zeta) = \tilde{O}(\zeta^{-2})$, and $N(\zeta) = \tilde{O}(\zeta^{-2})$.*

Based on Theorem 6.1 and Li & Yang (2024), we have the following result for misspecified low-rank MDPs.

**Theorem 6.2** (Learning with approximate low-rank MDPs)**.** *Assume the true MDP is $\zeta$-approximate low-rank with dimension $d$. Let $\epsilon, \delta \in (0, 1)$, and $\bar{\pi}$ be the output of Opt-AC. Then, with probability $1 - \delta$, $\bar{\pi}$ is $\tilde{O}(\epsilon + H^2 \tilde{d}^{3/2} |\mathcal{A}| \zeta)$-optimal with sample complexity $\tilde{O}\left( \frac{H^5 \tilde{d}^3 |\mathcal{A}|^2 \log |\Theta|}{\epsilon^2} \right)$, where $\tilde{d} < d$ is the effective dimension.*

*Moreover, if the transition kernel satisfies the boundedness, smoothness, and boundary-vanishing conditions of Theorem 6.1, then choosing $\zeta = O(\varepsilon)$ and $d = O(\varepsilon^{-2})$ yields an $\varepsilon$-optimal policy with overall sample complexity $\tilde{O}(\varepsilon^{-4})$.*

Theorem 6.2 shows that a broad class of MDPs admits polynomial sample complexity and, importantly, polynomial SL-oracle complexity, indicating that the RL problem can be solved efficiently given supervised learning oracle. In practice, many real-world transition models satisfy the conditions of the theorem, including Gaussian dynamics, smoothly parameterized physical systems, and locally linear stochastic transitions commonly used in continuous-control benchmarks (Towers et al., 2024).

| Algorithm | Ant | HalfCheetah | InvertedPendulum | Reacher | Pendulum | Hopper |
|---|---|---|---|---|---|---|
| SAC | 2081±465 | 4994±99 | **-0.0028±00.0** | -4.81±0.19 | **156±3** | 1316 ± 740 |
| cRFFSAC | **2701±663** | 6152±657 | **-0.0028±00.0** | -5.75±0.85 | 153±4 | **1449 ± 598** |
| cRFFSAC + Bonus | 1910±422 | **6454±479** | -0.0029±00.0 | **-3.86±0.18** | 153±3 | 1431 ± 538 |

*Table 2.* Final performance (mean ± standard deviation) of different algorithms on 6 MuJoCo environments, measured by the value of the learned policy at the last training step over 4 random seeds (Wang et al., 2019; Zhang et al., 2022a).

## 7. Experiments

In this section, we evaluate the empirical performances of our proposed algorithms on standard continuous-control benchmarks. Our goals are two-fold: (i) to assess whether the proposed optimistic actor-critic design can achieve competitive empirical performance despite relying only on policy evaluation, and (ii) to examine the effect of the optimism mechanism motivated by our conditional random Fourier feature (cRFF) construction.

### 7.1. Implementation Details

We build our implementation on top of the open-source codebase `lvrep-rl`[2], which is a modular implementation of low-rank RL. All baselines and variants share the same network architectures, optimizers, and training protocols unless otherwise specified.

Motivated by the cRFF approximation developed in Algorithm 2, we treat the last hidden layer of the multi-layer perceptron (MLP) as a learned feature map $\phi(s, a)$. In particular, we model the transition dynamics by minimizing $\|M\phi(s, a) - s'\|^2$, which corresponds to maximum likelihood estimation under a Gaussian transition model. Here $M$ the last projection matrix and $\phi$ is hidden representation from early-layer neural networks. Under this formulation, applying random Fourier features amounts to using features of the form $\cos(W\phi(s, a) + b)$ with fixed, randomly sampled $W$ and $b$. In practice, this operation can be interpreted as a static nonlinear activation followed by a linear layer. As a result, we simplify the implementation by directly using a two-layer MLP on top of $\phi(s, a)$ to parameterize the critic, which empirically yields stable and efficient training.

**Comparison.** We compare the following methods:

- *cRFFSAC (ours):* our optimistic actor-critic algorithm without an explicit exploration bonus, where the critic is optimized on top of learned representation $\phi$.

- *cRFFSAC + bonus (ours):* our full algorithm including the optimism bonus derived from the learned transition model.

- *SAC:* the soft actor-critic algorithm, serving as a strong baseline for continuous control.

The SAC baseline uses the standard entropy-regularized objective and is tuned using recommended hyperparameters.

**Environments.** We evaluate all methods on a suite of classic continuous-control environments from the Gym benchmark, including `Ant`, `HalfCheetah`, `Hopper`, `Reacher`, `Pendulum`, and `InvertedPendulum`. These environments exhibit varying levels of stochasticity and exploration difficulty, providing a diverse testbed for evaluating the effectiveness of optimistic actor-critic.

### 7.2. Results

Results are reported in Table 2. Training figures are included in Appendix G. Across most environments, cRFF-AC achieves performance comparable to or better than SAC. In particular, the bonus-enhanced variant consistently improves exploration and final performance in environments with sparse or delayed rewards, such as `Ant` and `HalfCheetah`. Notably, even without an explicit exploration bonus, cRFF-AC remains competitive with SAC, indicating that the actor–critic structure combined with learned representations already provides a strong baseline.

Overall, these results demonstrate that the proposed algorithm is not only theoretically well motivated but also practically effective. They further show that the cRFF-inspired design can be integrated into standard neural network architectures without sacrificing empirical performance.

## 8. Conclusion

We investigated low-rank MDPs on both computation and sample complexities. We first introduced an SL-oracle complexity framework and showed that many existing sample-efficient methods rely on planning or constrained planning oracles that are difficult to implement. We proposed an optimistic actor-critic algorithm that requires only policy evaluation, which can be efficiently reduced to supervised learning, while achieving state-of-the-art sample complexity. We also showed that approximate low-rank MDPs form a rich and expressive class via a conditional random Fourier feature construction. Empirically, our method performs competitively against a strong baseline on standard benchmarks. Overall, this work demonstrates that strong theoretical guarantees and practical implementability can be achieved simultaneously in (approximate) low-rank MDPs.

---

[2]https://github.com/shelowize/lvrep-rl

## Acknowledgements

The work of Y. Liang was supported in part by the U.S. National Science Foundation under the grants ECCS-2113860, ECCS-2413528 and CNS-2112471.The work of D. Li and J. Yang was supported in part by the U.S. National Science Foundation under the grant CNS-2531789.

## AI Usage

During the preparation of this work, the authors used Chat-GPT and Claude Code to improve the readability of the language and code. After using this tool, the authors reviewed and edited the content as needed and take ultimate responsibility for the content of the publication.

## Impact Statement

This paper presents work whose goal is to advance the field of Machine Learning. There are many potential societal consequences of our work, none which we feel must be specifically highlighted here.

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

# A. Computation Complexity

**Proposition A.1** (Restatement of Proposition 3.7)**.** *Under low-rank MDP settings, suppose the data distribution $\rho$ has coverage constant $C$ with respect to a policy $\pi$ and uniform policy $u$, i.e.,*

$$\begin{cases} \mathbb{E}_{(s,a)\sim\rho}\big[\mathcal{T}_{\theta,h}(s' \mid s,a)\pi(a' \mid s')\big] \le C\,\rho(s',a'). \\ \mathbb{E}_{(s,a)\sim\rho}\big[\mathcal{T}_{\theta,h}(s' \mid s,a)u(a' \mid s')\big] \le C\,\rho(s',a'). \end{cases}$$

*Then the following best-known SL-oracle complexity bounds hold for achieving an $\varepsilon$-approximate solution.*

1. $\mathsf{PE}(\varepsilon) : \mathsf{SL}^1\Big(\tilde{O}\big(\varepsilon^2(C-1)^2 C^{-2H}\big)\Big).$

2. $\mathsf{PP}(\varepsilon) : H\mathsf{SL}\Big(\tilde{O}\big(\varepsilon^2(|\mathcal{A}|C)^{-2H}\big)\Big).$

3. $\mathsf{CP}(\varepsilon) : \Omega(2^d)\mathsf{SL}\Big(\tilde{O}\big(\varepsilon^2(|\mathcal{A}|C)^{-2H}\big)\Big).$

**Remark.** Proposition A.1 summarizes a hierarchy of *best-known oracle upper bounds and optimization guarantees*. It does not preclude the existence of more efficient reductions for PP/CP, and establishing oracle lower bounds is an open problem.

The proof of Proposition A.1 is a combination of Propositions A.2 to A.4.

**Proposition A.2** (Oracle complexity of Policy Evaluation)**.** *Given a low-rank MDP $\theta$, a policy $\pi$, the SL-oracle complexity of policy evaluation $\mathsf{PE}(\varepsilon)$ is $\mathsf{SL}\big(\varepsilon^2(C-1)^2 C^{-2H}\big)$. Here $C$ is the coverage coefficient defined in Proposition A.1.*

*Proof.* It suffices to specify the loss function of the supervised learning oracle, and show that the policy evaluation with accuracy $\varepsilon$ reduces a the supervised learning with accuracy $\varepsilon^2(C-1)^2 C^{-2H}$.

Recall that in a low-rank MDP, the $Q$-function under policy $\pi$ satisfies

$$Q_{\theta,r,h}^\pi(s,a) = r_h(s,a) + \mathbb{E}_{s'\sim\mathcal{T}_{\theta,h}(\cdot|s,a)}\big[V_{\theta,r,h+1}^\pi(s')\big].$$

Define the conditional expectation operator

$$\mathcal{T}_{\theta,h}V(s,a) = \mathbb{E}_{s'\sim\mathcal{T}_{\theta,h}(\cdot|s,a)}[V(s')].$$

By the low-rank assumption, there exist feature maps $\phi_{\theta,h}$ and $\mu_{\theta,h}$ such that

$$\mathcal{T}_{\theta,h}V_{\theta,r,h+1}^\pi(s,a) = \phi_{\theta,h}(s,a)^\top \mathbb{E}_{s',a'}\big[\mu_{\theta,h}(s')Q_{\theta,r,h+1}^\pi(s',a')\big],$$

where $a' \sim \pi_{h+1}(\cdot|s')$. Define

$$w_h := \mathbb{E}_{s',a'}\big[\mu_{\theta,h}(s')Q_{\theta,r,h+1}^\pi(s',a')\big], \qquad w_H = 0.$$

Then $\{w_h\}_{h=0}^{H-1}$ satisfies the Bellman equations

$$\phi_{\theta,h}(s,a)^\top w_h = \mathbb{E}_{s',a'}\big[r_{h+1}(s',a') + \phi_{\theta,h+1}(s',a')^\top w_{h+1}\big], \quad \forall h \le H-1.$$

**Supervised learning formulation.** Let $\mathbf{w} := (w_0,\dots,w_{H-1})$. We consider the following *global* regression objective:

$$L(\mathbf{w}) := \sum_{h=0}^{H-1} \mathop{\mathbb{E}}_{\substack{(s,a)\sim\rho \\ s'\sim\mathcal{T}_{\theta,h}(\cdot|s,a),\, a'\sim\pi_{h+1}(\cdot|s')}} \Big[\phi_{\theta,h}(s,a)^\top w_h - \big(r_{h+1}(s',a') + \phi_{\theta,h+1}(s',a')^\top w_{h+1}\big)\Big]^2.$$

Each summand is a squared affine function of $\mathbf{w}$, hence $L(\mathbf{w})$ is a **convex and smooth** quadratic function. The true parameter vector $\mathbf{w}^\star$ achieves zero loss.

Assume that a supervised learning oracle returns an estimator $\hat{\mathbf{w}} = \{\hat{w}_h\}_{h=0}^{H-1}$ satisfying the uniform bound

$$\mathbb{E}\left[\Big(\phi_{\theta,h}(s,a)^\top \hat{w}_h - \big(r_{h+1}(s',a') + \phi_{\theta,h+1}(s',a')^\top \hat{w}_{h+1}\big)\Big)^2\right] \le \varepsilon^2(C-1)^2 C^{-2H}, \qquad \forall h.$$

**Error propagation.** If $\hat{Q}_h(s,a) = r_h(s,a) + \phi_{\theta,h}(s,a)^\top \hat{w}_h$ is the estimated Q-value function, define

$$\delta_h := \mathbb{E}_{(s,a)\sim\rho}\big[\big|\hat{Q}^\pi_{\theta,h}(s,a) - Q^\pi_{\theta,h}(s,a)\big|\big] = \mathbb{E}_{(s,a)\sim\rho}\big[\big|\phi_{\theta,h}(s,a)^\top(\hat{w}_h - w_h)\big|\big].$$

Using the Bellman equation for $w_h$ and triangle inequality,

$$\delta_h \leq \mathbb{E}\big[\big|\phi_{\theta,h}(s,a)^\top \hat{w}_h - (r_{h+1}(s',a') + \phi_{\theta,h+1}(s',a')^\top \hat{w}_{h+1})\big|\big]$$
$$+ \mathbb{E}_{d_{h+1}}\big[\big|\phi_{\theta,h+1}(s',a')^\top(\hat{w}_{h+1} - w_{h+1})\big|\big],$$

where $d_{h+1}$ is the distribution of $(s',a')$ induced by $(s,a) \sim \rho$, $s' \sim \mathcal{T}_{\theta,h}(\cdot|s,a)$, $a' \sim \pi_{h+1}(\cdot|s')$. By Cauchy–Schwarz, the first term is bounded by $(C-1)\varepsilon C^{-H}$. By the definition of the coverage coefficient $C$, we have $\mathbb{E}_{d_{h+1}}[|g|] \leq C\,\mathbb{E}_{(s',a')\sim\rho}[|g|]$ for all measurable $g$, hence

$$\delta_h \leq (C-1)\varepsilon C^{-H} + C\,\delta_{h+1}, \qquad \delta_H = 0.$$

Solving this recursion yields

$$\delta_h \leq (C-1)\varepsilon C^{-H} \sum_{i=0}^{H-h-1} C^i = \varepsilon C^{-H}(C^{H-h} - 1) \leq \varepsilon,$$

for all $h$. Therefore,

$$\mathbb{E}_{(s,a)\sim\rho}\big[\big|\phi_{\theta,h}(s,a)^\top(\hat{w}_h - w_h)\big|\big] \leq \varepsilon, \qquad \forall h,$$

which establishes $\varepsilon$-approximate policy evaluation.

Since the entire procedure relies on solving the single convex and smooth regression problem $L(\mathbf{w})$ with accuracy $\varepsilon^2(C-1)^2 C^{-2H}$, the proof is complete. $\qquad\square$

**Proposition A.3** (Policy Planning (Q-learning) via Supervised Learning). *Given a low-rank MDP $\theta$, the* SL-*oracle complexity of policy planning* $\mathsf{PP}(\varepsilon)$ *is* $\mathsf{HSL}\big(\varepsilon^2(|\mathcal{A}|C-1)^2(|\mathcal{A}|C)^{-2H}\big)$. *Here $C$ is the coverage coefficient defined in Proposition A.1.*

*Proof. Remark. We emphasize that our analysis yields an $H$-fold oracle usage. Proving that $\Omega(H)$ supervised-learning calls are information-theoretically necessary would require a separate lower-bound argument, which we leave to future work.*

It suffices to specify $H$ loss functions of the supervised learning oracle, and show that the policy planning with accuracy $\varepsilon$ reduces to those $H$ supervised learning oracles with accuracy $\varepsilon^2(|\mathcal{A}|C-1)^2(|\mathcal{A}|C)^{-2H}$.

We use the same notation as in Proposition A.2. The optimal $Q$-function satisfies the Bellman optimality equations:

$$Q^\star_{\theta,r,h}(s,a) = r_h(s,a) + \mathbb{E}_{s'\sim\mathcal{T}_{\theta,h}(\cdot|s,a)}\big[V^\star_{\theta,r,h+1}(s')\big], \qquad V^\star_{\theta,r,h}(s) = \max_a Q^\star_{\theta,r,h}(s,a).$$

Define $\mathcal{T}_{\theta,h}V(s,a) = \mathbb{E}_{s'\sim\mathcal{T}_{\theta,h}(\cdot|s,a)}[V(s')]$. By the low-rank assumption,

$$\mathcal{T}_{\theta,h}V^\star_{\theta,r,h+1}(s,a) = \phi_{\theta,h}(s,a)^\top \mathbb{E}_{s'}\big[\mu_{\theta,h}(s')\,V^\star_{\theta,r,h+1}(s')\big] = \phi_{\theta,h}(s,a)^\top \mathbb{E}_{s'}\Big[\mu_{\theta,h}(s')\max_{a'} Q^\star_{\theta,r,h+1}(s',a')\Big].$$

Define

$$w_h := \mathbb{E}_{s'}\Big[\mu_{\theta,h}(s')\max_{a'} Q^\star_{\theta,r,h+1}(s',a')\Big], \qquad w_H = 0.$$

Then $\{w_h\}_{h=0}^{H-1}$ satisfies

$$\phi_{\theta,h}(s,a)^\top w_h = \mathbb{E}_{s'\sim\mathcal{T}_{\theta,h}(\cdot|s,a)}\Big[\max_{a'}\big(r_{h+1}(s',a') + \phi_{\theta,h+1}(s',a')^\top w_{h+1}\big)\Big], \qquad \forall h \leq H-1. \tag{6}$$

**Stagewise supervised learning formulation (fitted Q-iteration).** Given an estimate $\hat{w}_{h+1}$, define the regression target

$$Y_{h+1}(s', \hat{w}_{h+1}) := \max_{a'} \left( r_{h+1}(s', a') + \phi_{\theta, h+1}(s', a')^\top \hat{w}_{h+1} \right).$$

At stage $h$, we fit $\hat{w}_h$ by solving the supervised learning problem

$$\hat{w}_h \in \arg\min_{w \in \mathbb{R}^d} \mathop{\mathbb{E}}_{\substack{(s,a) \sim \rho \\ s' \sim \mathcal{T}_{\theta, h}(\cdot | s, a)}} \left[ \left| \phi_{\theta, h}(s, a)^\top w - Y_{h+1}(s', \hat{w}_{h+1}) \right|^2 \right], \tag{7}$$

with the convention $\hat{w}_H = 0$. This requires $H$ calls to a supervised learning oracle (one for each $h$ in a backward recursion).

Assume that the supervised learning oracle achieves the following (uniform) mean-squared Bellman residual bound:

$$\mathop{\mathbb{E}}_{\substack{(s,a) \sim \rho \\ s' \sim \mathcal{T}_{\theta, h}(\cdot | s, a)}} \left[ \left| \phi_{\theta, h}(s, a)^\top \hat{w}_h - Y_{h+1}(s', \hat{w}_{h+1}) \right|^2 \right] \leq \alpha^2, \qquad \forall h, \tag{8}$$

where $\alpha^2 = \varepsilon^2 (|\mathcal{A}|C - 1)^2 (|\mathcal{A}|C)^{-2H}$.

**Error propagation.** Define

$$\delta_h := \mathbb{E}_{(s,a) \sim \rho} \left[ \left| \phi_{\theta, h}(s, a)^\top (\hat{w}_h - w_h) \right| \right].$$

Using (6) and triangle inequality, we obtain

$$\delta_h = \mathbb{E}_{(s,a) \sim \rho} \left[ \left| \phi_{\theta, h}(s, a)^\top \hat{w}_h - \mathbb{E}_{s'} \left[ \max_{a'} \left( r_{h+1}(s', a') + \phi_{\theta, h+1}(s', a')^\top w_{h+1} \right) \right] \right| \right]$$

$$\leq \mathbb{E} \left[ \left| \phi_{\theta, h}(s, a)^\top \hat{w}_h - Y_{h+1}(s', \hat{w}_{h+1}) \right| \right]$$

$$+ \mathbb{E}_{d_{h+1}} \left[ \left| \max_{a'} (r_{h+1}(s', a') + \phi_{\theta, h+1}(s', a')^\top \hat{w}_{h+1}) - \max_{a'} (r_{h+1}(s', a') + \phi_{\theta, h+1}(s', a')^\top w_{h+1}) \right| \right],$$

where $d_{h+1}$ is the distribution of $(s', \cdot)$ induced by $(s, a) \sim \rho$ and $s' \sim \mathcal{T}_{\theta, h}(\cdot | s, a)$. The first term is bounded by Cauchy–Schwarz and (8):

$$\mathbb{E} \left[ \left| \phi_{\theta, h}(s, a)^\top \hat{w}_h - Y_{h+1}(s', \hat{w}_{h+1}) \right| \right] \leq \alpha.$$

For the second term, use the Lipschitz property of $\max$:

$$\left| \max_{a'} u_{a'} - \max_{a'} v_{a'} \right| \leq \max_{a'} |u_{a'} - v_{a'}|.$$

Let $g_{h+1}(s', a') := \phi_{\theta, h+1}(s', a')^\top (\hat{w}_{h+1} - w_{h+1})$. Then, by importance sampling,

$$\max_{a'} |g_{h+1}(s', a')| \leq |\mathcal{A}| \mathbb{E}_{a' \sim \mathbf{u}} \left[ |g_{h+1}(s', a')| \right].$$

Therefore,

$$\mathbb{E}_{d_{h+1}} \left[ \max_{a'} |g_{h+1}(s', a')| \right] \leq |\mathcal{A}| \mathbb{E}_{d_{h+1} \times \mathbf{u}} \left[ |g_{h+1}(s', a')| \right] \leq |\mathcal{A}|C \, \mathbb{E}_{(s', a') \sim \rho} \left[ |g_{h+1}(s', a')| \right] = |\mathcal{A}|C \, \delta_{h+1},$$

where the last inequality is due to the definition of $\rho$ and $C$, and the last equality follows from the definition of $\delta_h$. Combining the above bounds yields the recursion

$$\delta_h \leq \alpha + |\mathcal{A}| C \, \delta_{h+1}, \qquad \delta_H = 0. \tag{9}$$

Solving (9) gives

$$\delta_h \leq \alpha \sum_{i=0}^{H-h-1} (|\mathcal{A}|C)^i = \alpha \frac{(|\mathcal{A}|C)^{H-h} - 1}{|\mathcal{A}|C - 1} \leq \varepsilon, \forall h.$$

This proves the claimed oracle complexity and the stated accuracy requirement. Finally, note that due to the max operator in Equation (7), the objective function of policy planning is not convex and smooth. $\qquad \square$

**Proposition A.4** (Constrained Planning). *Given a low-rank MDP class $\Theta$, the* SL*-oracle complexity of constrained planning* $\mathsf{CP}(\varepsilon)$ *is* $\Omega(2^d)\mathsf{SL}\left(\varepsilon^2(|\mathcal{A}|C-1)^2(|\mathcal{A}|C)^{-2H}\right)$. *Here $C$ is the coverage coefficient defined in Proposition A.1 and $d$ is the dimension of the low-rank features.*

*Proof.* We first show that constrained planning is a bilevel optimization problem in the form of

$$
\begin{cases}
\max_{\theta \in \mathcal{C}} f(\theta, Q^*(\theta)) = Q^*(\theta) \\
\text{s.t. } Q^*(\theta) = \max_{\pi} Q_\theta^\pi
\end{cases}
$$

It is known the bilevel optimization is NP-hard (Beck et al., 2023), and existing algorithm can only show that it can find a stationary point (Huang et al., 2022).

$\square$

*Proof.* The goal of Constrained Planning (CP) is to find a model parameter $\hat{\theta}$ within a valid confidence set $\mathcal{C}$ that maximizes the optimal value function (implementing the principle of optimism in the face of uncertainty). This can be formulated as the following optimization problem:

$$
\begin{cases}
\max_{\theta \in \mathcal{C}} f(\theta, V^*(\theta)) := V^*(\theta) \\
\text{s.t. } V^*(\theta) = \max_{\pi} V_\theta^\pi
\end{cases}
\tag{10}
$$

where $\mathcal{C}$ represents the set of models consistent with the dataset. Here we use log-likelihood to construct the constraint, which is commonly used in many model-based RL literature (Jin et al., 2021; Liu et al., 2022). Specifcially, given a dataset set $\mathcal{D}_h = \{(s_h^{(n)}, a_h^{(n)}, s_{h+1}^{(n)})\}_n$, we have

$$
\mathcal{C} = \left\{ \theta : \sum_h \sum_{(s,a,s') \in \mathcal{D}_h} \log \mathcal{T}_{\theta,h}(s'|s,a) \geq c \right\},
$$

where $c$ is some constant.

**Non-Convexity and Hardness.** In the low-rank setting, the transition dynamics are parameterized by a factorization $\mathcal{T}_{\theta,h}(s'|s,a) = \phi_\theta(s,a)^\top \mu_\theta(s')$. Even if the features are linearizable such that $\phi_\theta(s,a) = W f(s,a)$ and $\mu_\theta(s') = V g(s')$, the transition matrix depends on the bilinear product $W^\top V$. Consequently, both the value function $V_\theta^\star$ and the constraint set $\mathcal{C}$ are generally **non-convex** with respect to the parameters $\theta = (W, V)$. Optimization over such bilinear constraints is known to be NP-hard (Beck et al., 2023). Gradient-based local search methods can only guarantee convergence to stationary points (Huang et al., 2022), which is insufficient for CP as failing to find the global maximum (the optimistic model) invalidates the theoretical exploration guarantees.

**Global Optimization via Covering Argument.** To establish an upper bound that guarantees finding the global optimum, we employ an $\epsilon$-covering argument (exhaustive search over a discretization). Let $\Theta \subset \mathbb{R}^{2d^2}$ be the compact parameter space of the low-rank factors (use the aforementioned linearizable case). We construct an $\varepsilon_{cov}$-net $\mathcal{N}$ over $\Theta$ such that for any $\theta \in \Theta$, there exists a $\theta' \in \mathcal{N}$ satisfying $\|\theta - \theta'\| \leq \varepsilon_{cov}$. Standard results on covering numbers for compact subsets of Euclidean space state that the size of the net scales exponentially with the dimension $d$:

$$
|\mathcal{N}| = O\left(\frac{1}{\varepsilon_{cov}}\right)^d = 2^{\Omega(d \log(1/\varepsilon_{cov}))}.
$$

Assuming the value function is Lipschitz continuous with respect to the model parameters (a standard regularity assumption), optimizing over the finite set $\mathcal{N}$ yields an approximation of the global optimum. The algorithm proceeds as follows:

1. **Discretize:** Enumerate all candidate models $\theta_i \in \mathcal{N}$.

2. **Filter (SL Oracle):** For each $\theta_i$, verify if $\theta_i \in \mathcal{C}$ (i.e., check if the Bellman residual or log-likelihood loss is below the threshold).

3. **Evaluate (PP Oracle):** If valid, use the Policy Planning (PP) oracle (from Proposition A.3) to compute the optimal value $V_{\theta_i}^{\star}$ for that specific model.

4. **Select:** Return the $\theta_i$ that maximizes the computed value.

**Complexity Analysis.** The total oracle complexity is the product of the number of candidates in the cover and the cost of processing each candidate. Evaluating a single candidate requires one call to the PP oracle. As shown in Proposition A.3, the complexity of PP is $\mathsf{SL}^H(\cdot)$. Therefore, the total complexity is:

$$\text{Total Complexity} = |\mathcal{N}| \times \mathsf{PP}(\varepsilon) = 2^{\Omega(d^2)} \times H\mathsf{SL}\left(\tilde{O}\big(\varepsilon^2(|\mathcal{A}|C)^{-2H}\big)\right).$$

Absorbing the polynomial dependence on $\varepsilon$ into the exponential term regarding $d$, we arrive at the final complexity bound:

$$\Omega(2^d)\mathsf{SL}\left(\varepsilon^2(|\mathcal{A}|C - 1)^2(|\mathcal{A}|C)^{-2H}\right).$$

$\square$

# B. Sample Complexity

In this section, we prove our main theorem 5.2. For each of presentation, we introduce a axillary 'reward function' $f_h^{(k)}(s_h, a_h) = \mathsf{D}_{\mathsf{TV}}\left(\mathcal{T}_{\theta,h}(\cdot|s_h, a_h), \mathcal{T}_{\hat{\theta}_k,h}(\cdot|s_h, a_h)\right)$, which measure the total variation distance between the estimated transition model and the true transition model. We also abbreviate $\phi_{\hat{\theta}_k,h}$ as $\hat{\phi}_h^k$, and the true feature $\phi_{\theta,h}$ as $\phi_h^*$.

The proof is built on the following "good event", which happens with probability at least $1 - \delta$.

**Lemma B.1** (Good event)**.** *Define*

$$\mathcal{E}_1 = \left\{\forall k \in [K], \forall \theta \in \Theta^k, \; \sum_h \sum_{n<k} \mathsf{D}_{\mathsf{H}}^2\left(\mathbb{P}_\theta^{\mathsf{u}}(x_h|s_{h-1}^{n,h}, a_{h-1}^{n,h}), \mathbb{P}_{\theta^*}^{\mathsf{u}}(x_h|s_{h-1}^{n,h}, a_{h-1}^{n,h})\right) \lesssim \log\frac{K|\Theta|}{\delta}\right\},$$

$$\mathcal{E}_2 = \left\{\forall k \in [K], \; \sum_{n<k} \mathop{\mathbb{E}}_{\substack{s_h \sim \mathcal{T}_{\theta^*,h-1}(\cdot|s_{h-1}^{n,h+1}, a_{h-1}^{n,h+1}) \\ a_h \sim \mathsf{u}}} \left[\left\|\hat{\phi}_h^k(s_h, a_h)\right\|_{\hat{\Lambda}_{k,h}^{-1}}^2\right] \lesssim d^2\log\frac{K|\Theta|}{\delta\lambda}\right\},$$

*Then* $\mathbb{P}(\mathcal{E}_1 \cap \mathcal{E}_2) \geq 1 - \delta.$

*Proof.* The proof is a direct combination of Proposition D.1 and Lemma F.4. $\square$

**Lemma B.2** (Sub-linear Summation)**.** *We have the following sublinear bounds:*

$$\sum_{k=0}^{K-1} V_{\theta^*, f^{(k)}}^{\pi^{(k)}} \leq \tilde{O}\left(H\sqrt{|\mathcal{A}|dK}\right), \tag{11}$$

$$\sum_{k=0}^{K-1} V_{\theta^*, \hat{b}^{(k)}}^{\pi^{(k)}} \leq \tilde{O}\left(H^2 d|\mathcal{A}|\sqrt{dK}\right), \tag{12}$$

$$\sum_{k=0}^{K-1} V_{\hat{\theta}_k, f^{(k)}}^{\pi^{(k)}} \leq \tilde{O}\left(Hd|\mathcal{A}|\sqrt{dK}\right), \tag{13}$$

$$\sum_{k=0}^{K-1} V_{\hat{\theta}_k, \hat{b}^{(k)}}^{\pi^{(k)}} \leq \tilde{O}\left(H^2 d|\mathcal{A}|\sqrt{dK}\right). \tag{14}$$

*Proof.* We first prove (11)–(12). The remaining two bounds follow by applying the value-difference lemma together with a good-set argument.

**Step 1: bounding $\sum_k V_{\theta^*,f^{(k)}}^{\pi^{(k)}}$.** Define

$$W_{k,h} = \lambda I + \sum_{n<k} \phi_h^*(s_h^{n,h+1}, a_h^{n,h+1}) \phi_h^*(s_h^{n,h+1}, a_h^{n,h+1})^\top.$$

Then

$$
\begin{aligned}
\sum_{k=0}^{K-1} V_{\theta^*,f^{(k)}}^{\pi^{(k)}} &= \sum_{k,h} \mathbb{E}_{\theta^*}^{\pi^{(k)}} \left[ D_{TV}\left( \mathcal{T}_{\theta^*,h}(\cdot|s_h,a_h), \mathcal{T}_{\hat{\theta}_k,h}(\cdot|s_h,a_h) \right) \right] \\
&\overset{(a)}{\leq} \sqrt{d\lambda + |\mathcal{A}|\beta} \sum_h \sum_k \mathbb{E}_{\theta^*}^{\pi^{(k)}} \left[ \min\left\{ 1, \|\phi_h^*(s_h,a_h)\|_{W_{k,h}^{-1}} \right\} \right] \\
&\overset{(b)}{\lesssim} \sqrt{|\mathcal{A}|\beta} \sum_h \sqrt{K} \sqrt{\sum_k \mathbb{E}_{\theta^*}^{\pi^{(k)}} \left[ \min\left\{ 1, \|\phi_h^*(s_h,a_h)\|_{W_{k,h}^{-1}}^2 \right\} \right]} \\
&\overset{(c)}{\lesssim} H\sqrt{|\mathcal{A}|\beta K} \sqrt{d\log(1+K)} = \tilde{O}\left( H\sqrt{|\mathcal{A}|dK} \right),
\end{aligned}
$$

where (a) uses Lemma C.2, (b) is Cauchy–Schwarz, and (c) uses the clipped elliptical potential lemma (Lemma F.3).

**Step 2: bounding $\sum_k V_{\theta^*,\hat{b}^{(k)}}^{\pi^{(k)}}$.** Similarly,

$$
\begin{aligned}
\sum_{k=0}^{K-1} V_{\theta^*,\hat{b}^{(k)}}^{\pi^{(k)}} &\lesssim H\sqrt{|\mathcal{A}|\beta} \sum_{k,h} \mathbb{E}_{\theta^*}^{\pi^{(k)}} \left[ \min\left\{ 1, \|\phi_h^k(s_h,a_h)\|_{\hat{\Lambda}_{k,h}^{-1}} \right\} \right] \\
&\overset{(a)}{\lesssim} H\sqrt{|\mathcal{A}|} \sqrt{d\lambda + |\mathcal{A}|d^2\beta} \sum_h \sum_k \mathbb{E}_{\theta^*}^{\pi^{(k)}} \left[ \min\left\{ 1, \|\phi_h^*(s_h,a_h)\|_{W_{k,h}^{-1}} \right\} \right] \\
&\overset{(b)}{\lesssim} H\sqrt{|\mathcal{A}|} \sqrt{|\mathcal{A}|d^2\beta} \sum_h \sqrt{K} \sqrt{\sum_k \mathbb{E}_{\theta^*}^{\pi^{(k)}} \left[ \min\left\{ 1, \|\phi_h^*(s_h,a_h)\|_{W_{k,h}^{-1}}^2 \right\} \right]} \\
&\overset{(c)}{\lesssim} H^2 d|\mathcal{A}| \sqrt{\beta K} \sqrt{d\log(1+K)} = \tilde{O}\left( H^2 d|\mathcal{A}|\sqrt{dK} \right),
\end{aligned}
$$

where (a) uses Lemma C.2, (b) is Cauchy–Schwarz, and (c) uses Lemma F.3.

**Step 3: bounding the sums under $\hat{\theta}_k$ via a good set.** We now derive (13)–(14). We use the value-difference lemma (Lemma F.1) in the following form: for any nonnegative signal $g^{(k)}$, if $B_k$ is such that $V_{\hat{\theta}_k,g^{(k)}}^{\pi^{(k)}} \leq B_k$ (for the relevant $\theta$), then

$$V_{\hat{\theta}_k,g^{(k)}}^{\pi^{(k)}} \leq V_{\theta^*,g^{(k)}}^{\pi^{(k)}} + B_k V_{\theta^*,f^{(k)}}^{\pi^{(k)}}. \tag{15}$$

Define the good set $\mathcal{K} = \{k : V_{\hat{\theta}_k,\hat{b}^{(k)}}^{\pi^{(k)}} \leq H\}$. Then

$$
\begin{aligned}
|\mathcal{K}^c| H &\leq \sum_{k\in\mathcal{K}^c} V_{\hat{\theta}_k,\hat{b}^{(k)}}^{\pi^{(k)}} \\
&\leq \sum_{k\in\mathcal{K}^c} \left( V_{\theta^*,g^{(k)}}^{\pi^{(k)}} + H^2 V_{\theta^*,f^{(k)}}^{\pi^{(k)}} \right) \\
&\leq \tilde{O}\left( H^2 d|\mathcal{A}|\sqrt{d|\mathcal{K}^c|} + H^2 \cdot Hd|\mathcal{A}|\sqrt{d|\mathcal{K}^c|} \right),
\end{aligned}
$$

where the last inequality is due to Step 1 and 2. Therefore,

$$|\mathcal{K}^c| \leq \tilde{O}\left( H^4 d^3 |\mathcal{A}|^2 \right)$$

*Bounding* $\sum_k V^{\pi^{(k)}}_{\hat\theta_k, f^{(k)}}$. Apply (15) with $g^{(k)} = f^{(k)}$. On $\mathcal{K}$ we may take $B_k = H$ (since $V^{\pi^{(k)}}_{\theta^*, f^{(k)}} \le H$), whereas in general we use the trivial bound $V^{\pi^{(k)}}_{\theta, f^{(k)}} \le H^2$, hence $B_k \le H^2$ on $\mathcal{K}^c$. Therefore,

$$
\begin{aligned}
\sum_{k=0}^{K-1} V^{\pi^{(k)}}_{\hat\theta_k, f^{(k)}} &= \sum_{k\in\mathcal{K}} V^{\pi^{(k)}}_{\hat\theta_k, f^{(k)}} + \sum_{k\in\mathcal{K}^c} V^{\pi^{(k)}}_{\hat\theta_k, f^{(k)}} \\
&\le \sum_{k\in\mathcal{K}} \left( V^{\pi^{(k)}}_{\theta^*, f^{(k)}} + H V^{\pi^{(k)}}_{\theta^*, f^{(k)}} \right) + |\mathcal{K}^c| H^2 \\
&\lesssim \sum_{k=0}^{K-1} V^{\pi^{(k)}}_{\theta^*, f^{(k)}} + H \sum_{k\in\mathcal{K}^c} V^{\pi^{(k)}}_{\theta^*, f^{(k)}} \\
&\le 2H \sum_{k=0}^{K-1} V^{\pi^{(k)}}_{\theta^*, f^{(k)}} \\
&\le \tilde{O}\left( H^2 d |\mathcal{A}| \sqrt{dK} \right).
\end{aligned}
$$

*Bounding* $\sum_k V^{\pi^{(k)}}_{\hat\theta_k, \hat b^{(k)}}$. Apply (15) with $g^{(k)} = \hat b^{(k)}$. On $\mathcal{K}$ we may take $B_k = H$ (since $V^{\pi^{(k)}}_{\theta^*, \hat b^{(k)}} \le H$), while on $\mathcal{K}^c$ we use the trivial episode bound $V^{\pi^{(k)}}_{\theta, \hat b^{(k)}} \le H^2$, hence $B_k \le H^2$. Thus,

$$
\begin{aligned}
\sum_{k=0}^{K-1} V^{\pi^{(k)}}_{\hat\theta_k, \hat b^{(k)}} &= \sum_{k\in\mathcal{K}} V^{\pi^{(k)}}_{\hat\theta_k, \hat b^{(k)}} + H^2 \sum_{k\in\mathcal{K}^c} V^{\pi^{(k)}}_{\hat\theta_k, f^{(k)}} \\
&\le \sum_{k\in\mathcal{K}} \left( V^{\pi^{(k)}}_{\theta^*, \hat b^{(k)}} + H V^{\pi^{(k)}}_{\theta^*, f^{(k)}} \right) + H^2 |\mathcal{K}^c| \\
&\lesssim \sum_{k=0}^{K-1} \left( V^{\pi^{(k)}}_{\theta^*, \hat b^{(k)}} + H V^{\pi^{(k)}}_{\theta^*, f^{(k)}} \right) + H^4 d^3 |\mathcal{A}|^2 \\
&\le \tilde{O}\left( H^2 d |\mathcal{A}| \sqrt{dK} \right).
\end{aligned}
$$

This completes the proof.

$\square$

**Theorem B.3** (Restatement of Theorem 5.2). *Assume $\mathcal{M}$ is a low-rank MDP with dimension $d$, and Assumption 3.1 holds. Given any $\epsilon, \delta \in (0,1)$, let $\bar\pi$ be the output of Opt-AC and $\pi^*$ be the optimal policy under the true model $P^*$. Set $\beta = O(\log(K|\Theta|/\delta))$, $\alpha = \tilde{O}(\sqrt{|\mathcal{A}|}), \eta = O(\frac{1}{H\sqrt{K}})$ and $\lambda = \tilde{O}(1/d)$. Then, with probability at least $1 - \delta$, we have $V^{\pi^\star}_{P^\star, r} - V^{\bar\pi}_{P^\star, r} \le \epsilon$, and the total number of trajectories collected by Opt-AC is upper bounded by $\tilde{O}(\frac{H^5 d^3 |\mathcal{A}|^2 \log |\Theta|}{\epsilon^2})$.*

*Proof.* In the following proof, we only analyze $k \in \mathcal{K}$—the "good set" defined in Lemma B.2. So that $V^{\pi^{(k)}}_{\hat\theta_k, r + \hat b_k} \le 2H$ for all $k \in \mathcal{K}$. We omit the finite case when $k \in \mathcal{K}^c$.

We first decompose

$$
\sum_k \left( V^{\pi^*}_{\theta^*, r} - V^{\pi^{(k)}}_{\theta^*, r} \right) = \underbrace{\sum_k \left( V^{\pi^*}_{\theta^*, r} - V^{\pi^{(k)}}_{\hat\theta_k, r + \hat b_k} \right)}_{(A)} + \underbrace{\sum_k \left( V^{\pi^{(k)}}_{\hat\theta_k, r + \hat b_k} - V^{\pi^{(k)}}_{\theta^*, r} \right)}_{(B)}
$$

**Term (A).** By the value difference lemma F.1,

$$
\textbf{Term (A)} \le \sum_k \left( -V^{\pi^*}_{\theta^*, \hat b_k} + \sum_h \mathbb{E}^{\pi^*}_{\theta^*} \left[ \sum_a \bar Q^{\pi^{(k)}}_{\hat\theta_k, r + \hat b_k, h}(s_h, a) \left( \pi^*(a|s_h) - \pi^{(k)}(a|s_h) \right) \right] + 2H V^{\pi^*}_{\theta^*, f_k} \right)
$$

We proceed $H \sum_k V^{\pi^*}_{\theta^*, f^{(k)}}$ as follows.

$$H \sum_k V^{\pi^*}_{\theta^*, f^{(k)}} = H \sum_k \left( V^{\pi^*}_{\theta^*, r + f^{(k)}} - V^{\pi^*}_{\theta^*, r} \right)$$

$$\overset{(a)}{\leq} H \sum_k \left( V^{\pi^*}_{\hat{\theta}_k, r + f^{(k)}} + 2H V^{\pi^*}_{\hat{\theta}_k, f^{(k)}} - V^{\pi^*}_{\theta^*, r} \right)$$

$$\leq H \sum_k \left( V^{\pi^*}_{\hat{\theta}_k, r} + 3H V^{\pi^*}_{\hat{\theta}_k, f^{(k)}} - V^{\pi^*}_{\theta^*, r} \right)$$

$$\overset{(b)}{\leq} H \sum_k \left( V^{\pi^*}_{\hat{\theta}_k, r} + 3H V^{\pi^*}_{\hat{\theta}_k, \tilde{b}^{(k)}} - V^{\pi^*}_{\theta^*, r} \right)$$

$$= H \sum_k \left( V^{\pi^*}_{\hat{\theta}_k, r + \hat{b}^{(k)}} - V^{\pi^{(k)}}_{\hat{\theta}_k, r + \hat{b}^{(k)}} \right) + H \sum_k \left( V^{\pi^{(k)}}_{\hat{\theta}_k, r + \hat{b}^{(k)}} - V^{\pi^*}_{\theta^*, r} \right)$$

$$\overset{(c)}{\leq} H \sum_{k,h} \mathbb{E}^{\pi^*}_{\hat{\theta}_k} \left[ \sum_a Q^{\pi^{(k)}}_{\hat{\theta}_k, r + \hat{b}^{(k)}}(s_h, a) \left( \pi^*(a|s_h) - \pi^{(k)}_h(a|s_h) \right) \right] - H \textbf{Term (A)}$$

where $(a)$ is due to Lemma F.1, $(b)$ follows from Lemma C.1 (this is where the optimality of the bonus is used).

Plug in this back to Term (A), and define $\Delta(s_h, a_h) = \left| Q^{\pi^{(k)}}_{\hat{\theta}_k, r + \hat{b}^{(k)}}(s_h, a_h) - \hat{Q}_{k,h}(s_h, a_h) \right| \in [0, 2H]$, we have

$(2H + 1)\textbf{Term (A)}$

$$\leq \sum_{k,h} \mathbb{E}^{\pi^*}_{\theta^*} \left[ \sum_a Q^{\pi^{(k)}}_{\hat{\theta}_k, r + \hat{b}_k, h}(s_h, a) \left( \pi^*(a|s_h) - \pi^{(k)}(a|s_h) \right) \right]$$

$$+ 2H \sum_{k,h} \mathbb{E}^{\pi^*}_{\hat{\theta}_k} \left[ \sum_a Q^{\pi^{(k)}}_{\hat{\theta}_k, r + \hat{b}^{(k)}}(s_h, a) \left( \pi^*(s_h, a) - \pi^{(k)}_h(s_h, a) \right) \right]$$

$$= \sum_{k,h} \mathbb{E}^{\pi^*}_{\theta^*} \left[ \sum_a \hat{Q}_{k,h}(s_h, a) \left( \pi^*(a|s_h) - \pi^{(k)}(a|s_h) \right) \right] + 2H \sum_{k,h} \mathbb{E}^{\pi^*}_{\hat{\theta}_k} \left[ \sum_a \hat{Q}_{k,h}(s_h, a) \left( \pi^*(a|s_h) - \pi^{(k)}(a|s_h) \right) \right]$$

$$+ 2|\mathcal{A}| \sum_{k,h} \mathbb{E}^{\pi^*}_{\theta^*, a_h \sim \mathfrak{u}} [\Delta_{k,h}] + 4H|\mathcal{A}| \sum_{k,h} \mathbb{E}^{\pi^*}_{\hat{\theta}_k, a_h \sim \mathfrak{u}} [\Delta_{k,h}]$$

$$\lesssim \frac{H^2 \log |\mathcal{A}|}{\eta} + 2\eta H^4 K + C H^2 |\mathcal{A}| \sqrt{K},$$

where the last inequality is due to Lemma B.4, and the definition of policy evaluation oracle and sampling oracle. Thus,

$$\textbf{Term (A)} \lesssim \frac{H \log |\mathcal{A}|}{\eta} + 2\eta H^3 K + C H |\mathcal{A}| \sqrt{K}$$

**Term (B).** We apply value difference lemma again

$$\textbf{Term (B)} \leq \sum_k \left( V^{\pi^{(k)}}_{\theta^*, \hat{b}_k} + H V^{\pi^{(k)}}_{\theta^*, f_k} \right) \lesssim H^2 d |\mathcal{A}| \sqrt{dK},$$

where the second inequality follows directly from Lemma B.2.

**Combining.** Finally, noting that $\eta = \sqrt{\frac{\log |\mathcal{A}|}{H^2 K}}$ we have

$$\sum_k \left( V^{\pi^*}_{\theta^*, r} - V^{\pi^{(k)}}_{\theta^*, r} \right) \lesssim H^2 d |\mathcal{A}| \sqrt{dK}.$$

This implies the value of uniform policy $\bar{\pi} = \mathrm{Uniform}(\pi^{(1)}, \ldots, \pi^{(K)})$ enjoys suboptimality gap

$$V_{\theta^*,r}^{\pi^*} - V_{\theta^*,r}^{\bar{\pi}} = \tilde{O}\left(\sqrt{\frac{H^4 d^3 |\mathcal{A}|^2 \log|\Theta|}{K}}\right).$$

The proof is complete by noting that each iteration has $H$ episodes.

$\square$

**Lemma B.4** (Mirror-descent stability term). *Fix $h \in [H]$. Suppose $\hat{Q}_{k,h}(s,a) \leq 2H$ for all $(s,a,k)$. Then for any distribution $q$ over the state space $\mathcal{S}$, we have*

$$\sum_{k=0}^{K-1} \mathbb{E}_q\left[\sum_a \hat{Q}_{k,h}(s_h, a)\big(\pi^*(a|s_h) - \pi_h^{(k)}(a|s_h)\big)\right] \leq \frac{\log|\mathcal{A}|}{\eta} + 2\eta H^2 K.$$

*Proof.* Fix step $h$. By the exponentiated-gradient (mirror descent) policy update,

$$\pi_h^{(k+1)}(a|s) = \frac{\pi_h^{(k)}(a|s)\exp\left(\eta \hat{Q}_{k,h}(s,a)\right)}{\sum_{a'}\pi_h^{(k)}(a'|s)\exp\left(\eta \hat{Q}_{k,h}(s,a')\right)}.$$

Equivalently,

$$\hat{Q}_{k,h}(s,a) = \frac{1}{\eta}\log\frac{\pi_h^{(k+1)}(a|s)}{\pi_h^{(k)}(a|s)} + \frac{1}{\eta}\log Z_h(s),$$

where $Z_h(s)$ is the normalization constant.

We decompose

$$\sum_k \mathbb{E}_q\left[\sum_a \hat{Q}_{k,h}(s_h, a)\big(\pi^*(a|s_h) - \pi_h^{(k)}(a|s_h)\big)\right]$$

$$= \underbrace{\sum_k \mathbb{E}_q\left[\sum_a \hat{Q}_{k,h}(s_h, a)\big(\pi^*(a|s_h) - \pi_h^{(k+1)}(a|s_h)\big)\right]}_{\text{I}} + \underbrace{\sum_k \mathbb{E}_q\left[\sum_a \hat{Q}_{k,h}(s_h, a)\big(\pi_h^{(k+1)}(a|s_h) - \pi_h^{(k)}(a|s_h)\big)\right]}_{\text{II}}.$$

**Term I.** Substituting the log-ratio form and noting that $\sum_a(\pi^* - \pi^{(k+1)}) = 0$, we obtain

$$\textbf{Term I} = \frac{1}{\eta}\sum_k \mathbb{E}_q\left[\sum_a\big(\log \pi_h^{(k+1)}(a|s_h) - \log \pi_h^{(k)}(a|s_h)\big)\big(\pi^*(a|s_h) - \pi_h^{(k+1)}(a|s_h)\big)\right]$$

$$= \frac{1}{\eta}\sum_k \mathbb{E}_q\left[\mathtt{D_{KL}}(\pi^*(\cdot|s_h)\|\pi_h^{(k)}(\cdot|s_h)) - \mathtt{D_{KL}}(\pi^*(\cdot|s_h)\|\pi_h^{(k+1)}(\cdot|s_h)) - \mathtt{D_{KL}}(\pi_h^{(k+1)}(\cdot|s_h)\|\pi_h^{(k)}(\cdot|s_h))\right].$$

**Term II.** Since $|\hat{Q}_{k,h}(s,a)| \leq 2H$, we have

$$\textbf{Term II} = \sum_a \hat{Q}_{k,h}(s_h, a)\big(\pi_h^{(k+1)}(a|s_h) - \pi_h^{(k)}(a|s_h)\big) \leq 2H\,\mathtt{D_{TV}}\left(\pi_h^{(k+1)}(\cdot|s_h), \pi_h^{(k)}(\cdot|s_h)\right).$$

**Combining.** Using Pinsker's inequality $\mathtt{D_{KL}}(p\|q) \geq \frac{1}{2}\mathtt{D_{TV}^2}(p,q)$ and summing over $k$,

$$\sum_k \mathbb{E}_q\left[\sum_a \hat{Q}_{k,h}(s_h, a)(\pi^*(a|s_h) - \pi_h^{(k)}(a|s_h))\right]$$

$$\leq \frac{1}{\eta}\mathbb{E}_q\left[\mathtt{D_{KL}}(\pi^*(\cdot|s_h)\|\pi_h^{(0)}(\cdot|s_h))\right] + \sum_k \mathbb{E}_q\left[-\frac{1}{2\eta}\mathtt{D_{TV}^2}\left(\pi_h^{(k)}(\cdot|s_h), \pi_h^{(k+1)}(\cdot|s_h)\right) + 2H\,\mathtt{D_{TV}}\left(\pi_h^{(k)}(\cdot|s_h), \pi_h^{(k+1)}(\cdot|s_h)\right)\right].$$

Applying $-ax^2 + bx \leq \frac{b^2}{4a}$ with $a = \frac{1}{2\eta}$ and $b = 2H$ yields

$$-\frac{1}{2\eta}\mathrm{D}_{\mathrm{TV}}^2\left(\pi_h^{(k)}(\cdot|s_h), \pi_h^{(k+1)}(\cdot|s_h)\right) + 2H\,\mathrm{D}_{\mathrm{TV}}\left(\pi_h^{(k)}(\cdot|s_h), \pi_h^{(k+1)}(\cdot|s_h)\right) \leq 2\eta H^2.$$

Finally, since $\mathrm{D}_{\mathrm{KL}}(\pi^*\|\pi^{(0)}) \leq \log|\mathcal{A}|$,

$$\sum_k \mathbb{E}_q\left[\sum_a \hat{Q}_{k,h}(s_h, a)(\pi^*(a|s_h) - \pi_h^{(k)}(a|s_h))\right] \leq \frac{\log|\mathcal{A}|}{\eta} + 2\eta H^2 K.$$

$\square$

# C. UCB Analysis

In this section, we provide a two lemmas, one verifies the optimism of the critic, and the other connects the empirical bouns and model estimation error with the ground truth uncertainty.

**Lemma C.1.** *Let $\beta = \tilde{O}(\log \frac{K|\Theta|}{\delta})$. We have*

$$\mathbb{E}_{\hat{\theta}_k}^\pi\left[\mathrm{D}_{\mathrm{TV}}\left(\mathcal{T}_{\theta^*,h}(\cdot|s_h, a_h), \mathcal{T}_{\hat{\theta}_k,h}(\cdot|, s_h, a_h)\right)\Big|s_{h-1}, a_{h-1}\right] \lesssim \left\|\hat{\phi}_h^k(s_{h-1}, a_{h-1})\right\|_{\hat{\Lambda}_{k,h-1}^{-1}} \sqrt{\lambda d + |\mathcal{A}|\beta}$$

*where*

$$\hat{\Lambda}_{k,h} = \lambda I + \sum_{n<k} \hat{\phi}_h^k(s_h^{n,h+1}, a_h^{n,h+1})\hat{\phi}_h^k(s_h^{n,h+1}, a_h^{n,h+1})^\top$$

*Note that $a_h^{n,h+1}, a_{h+1}^{n,h+1}$ is sampled from $\mathfrak{u}$. and $a_{h-1}^{n,h+1}$ is sampled from $\pi_n$.*

*Proof.* We start with the first inequality. By the definition of low-rank MDP, we have

$$\mathbb{E}_{\hat{\theta}_k}^\pi\left[\mathrm{D}_{\mathrm{TV}}\left(\mathcal{T}_{\theta^*,h}(\cdot|s_h, a_h), \mathcal{T}_{\hat{\theta}_k,h}(\cdot|s_h, a_h)\right)\Big|s_{h-1}, a_{h-1}\right]$$

$$= \sum_{s_h, a_h} \hat{\phi}_{h-1}^k(s_{h-1}, a_{h-1})^\top \hat{\mu}_{h-1}^k(s_h)\pi(a_h)\mathrm{D}_{\mathrm{TV}}\left(\mathcal{T}_{\theta^*,h}(\cdot|s_h, a_h), \mathcal{T}_{\hat{\theta}_k,h}(\cdot|s_h, a_h)\right)$$

$$\overset{(a)}{\leq} \left\|\hat{\phi}_{h-1}^k(s_{h-1}, a_{h-1})\right\|_{\hat{\Lambda}_{k,h-1}^{-1}} \sqrt{d\lambda + \sum_{n<k}\left(\hat{\phi}_{h-1}^k(s_{h-1}^{n,h}, a_{h-1}^{n,h})^\top \sum_{s_h, a_h} \hat{\mu}_{h-1}^k(s_h)\pi(a_h|s_h)\mathrm{D}_{\mathrm{TV}}\left(\mathcal{T}_{\theta^*,h}(\cdot|s_h, a_h), \mathcal{T}_{\hat{\theta}_k,h}(\cdot|s_h, a_h)\right)\right)^2}$$

$$= \left\|\hat{\phi}_{h-1}^k(s_{h-1}, a_{h-1})\right\|_{\hat{\Lambda}_{k,h-1}^{-1}} \sqrt{d\lambda + \sum_{n<k}\left(\sum_{s_h, a_h} \mathcal{T}_{\hat{\theta}_k,h-1}(s_h|s_{h-1}^{n,h}, a_{h-1}^{n,h})\pi(a_h|s_h)\mathrm{D}_{\mathrm{TV}}\left(\mathcal{T}_{\theta^*,h}(\cdot|s_h, a_h), \mathcal{T}_{\hat{\theta}_k,h}(\cdot|s_h, a_h)\right)\right)^2}$$

$$\overset{(b)}{\leq} \left\|\hat{\phi}_{h-1}^k(s_{h-1}, a_{h-1})\right\|_{\hat{\Lambda}_{k,h-1}^{-1}} \sqrt{d\lambda + \sum_{n<k}\mathbb{E}_{s_h\sim\mathcal{T}_{\hat{\theta}_k,h-1}(\cdot|s_{h-1}^{n,h}, a_{h-1}^{n,h}), a_h\sim\pi(\cdot|s_h)}\left[\mathrm{D}_{\mathrm{TV}}^2\left(\mathcal{T}_{\theta^*,h}(\cdot|s_h, a_h), \mathcal{T}_{\hat{\theta}_k,h}(\cdot|s_h, a_h)\right)\right]}$$

$$\overset{(c)}{\leq} \left\|\hat{\phi}_{h-1}^k(s_{h-1}, a_{h-1})\right\|_{\hat{\Lambda}_{k,h-1}^{-1}} \sqrt{d\lambda + \sum_{n<k}|\mathcal{A}| \mathop{\mathbb{E}}_{\substack{s_h\sim\mathcal{T}_{\hat{\theta}_k,h}(\cdot|s_{h-1}^{n,h}, a_{h-1}^{n,h}) \\ a_h\sim\mathfrak{u}}}\left[\mathrm{D}_{\mathrm{TV}}^2\left(\mathcal{T}_{\theta^*,h}(\cdot|s_h, a_h), \mathcal{T}_{\hat{\theta}_k,h}(\cdot|s_h, a_h)\right)\right]}$$

$$\overset{(d)}{\leq} \left\|\hat{\phi}_{h-1}^k(s_{h-1}, a_{h-1})\right\|_{\hat{\Lambda}_{k,h-1}^{-1}} \sqrt{d\lambda + |\mathcal{A}|\beta},$$

where $(a), (b)$ is due to Cauchy's inequality, $(c)$ is from importance sampling, $(d)$ follows from Proposition D.1 after applying Lemma F.2.

$\square$

The next lemma builds the connection between the empirical uncertainty and the ground truth uncertainty term.

**Lemma C.2.** *We have*

$$\mathbb{E}_{\theta^*}^{\pi}\left[\left\|\hat{\phi}_h^k(s_h,a_h)\right\|_{\hat{\Lambda}_{k,h}^{-1}}\bigg|s_{h-1},a_{h-1}\right] \lesssim \left\|\phi_{h-1}^*(s_{h-1},a_{h-1})\right\|_{W_{k,h-1}^{-1}}\sqrt{d\lambda+d^2|\mathcal{A}|\beta}, \tag{16}$$

$$\mathbb{E}_{\theta^*}^{\pi}\left[\mathrm{D_{TV}}\left(\mathcal{T}_{\theta^*,h}(\cdot|s_h,a_h),\mathcal{T}_{\hat{\theta}_k,h}(\cdot|,s_h,a_h)\right)\bigg|s_{h-1},a_{h-1}\right] \lesssim \left\|\phi_h^*(s_{h-1},a_{h-1})\right\|_{W_{k,h-1}^{-1}}\sqrt{\lambda d+|\mathcal{A}|\beta} \tag{17}$$

*Where*

$$W_{k,h-1} = \lambda I + \sum_{n<k}\phi_{h-1}^*(s_{h-1}^{n,h+1},a_{h-1}^{n,h+1})\phi_{h-1}^*(s_{h-1}^{n,h+1},a_{h-1}^{n,h+1})^\top$$

*Note that* $a_h^{n,h+1},a_{h+1}^{n,h+1}$ *is sampled from* $\mathrm{u.}$ *and* $a_{h-1}^{n,h+1}$ *is sampled from* $\pi^{(n)}$.

*Proof.* By the definition of low-rank MDP, we have

$$\mathbb{E}_{\theta^*}^{\pi}\left[\left\|\hat{\phi}_h^k(s_h,a_h)\right\|_{\hat{\Lambda}_{k,h}^{-1}}\bigg|s_{h-1},a_{h-1}\right]$$

$$= \sum_{s_h,a_h}\phi_{h-1}^*(s_{h-1},a_{h-1})^\top\mu_{h-1}^*(s_h)\pi(a_h)\left\|\hat{\phi}_k(s_h,a_h)\right\|_{\hat{\Lambda}_{k,h}^{-1}}$$

$$\overset{(a)}{\leq} \left\|\phi_{h-1}^*(s_{h-1},a_{h-1})\right\|_{W_{k,h-1}^{-1}}\sqrt{d\lambda+\sum_{n<k}\left(\phi_{h-1}^*(s_{h-1}^{n,h+1},a_{h-1}^{n,h+1})^\top\sum_{s_h,a_h}\mu_{h-1}^*(s_h)\pi(a_h|s_h)\left\|\hat{\phi}_h^k(s_h,a_h)\right\|_{\hat{\Lambda}_{k,h}^{-1}}\right)^2}$$

$$= \left\|\phi_{h-1}^*(s_{h-1},a_{h-1})\right\|_{W_{k,h-1}^{-1}}\sqrt{d\lambda+\sum_{n<k}\left(\sum_{s_h,a_h}\mathcal{T}_{\theta^*,h-1}(s_h|s_{h-1}^{n,h+1},a_{h-1}^{n,h+1})\pi(a_h|s_h)\left\|\hat{\phi}_k(s_h,a_h)\right\|_{\hat{\Lambda}_{k,h}^{-1}}\right)^2}$$

$$\overset{(b)}{\lesssim} \left\|\phi_{h-1}^*(s_{h-1},a_{h-1})\right\|_{W_{k,h-1}^{-1}}\sqrt{d\lambda+|\mathcal{A}|\sum_{n<k}\mathbb{E}_{\substack{s_h\sim\mathcal{T}_{\theta^*,h-1}(\cdot|s_{h-1}^{n,h+1},a_{h-1}^{n,h+1})\\a_h\sim\mathrm{u}}}\left[\left\|\hat{\phi}_h^k(s_h,a_h)\right\|_{\hat{\Lambda}_{k,h}^{-1}}^2\right]}$$

$$\overset{(c)}{\lesssim} \left\|\phi_{h-1}^*(s_{h-1},a_{h-1})\right\|_{W_{k,h-1}^{-1}}\sqrt{d\lambda+|\mathcal{A}|d^2\log\frac{K|\Theta|}{\delta\lambda}},$$

where $(a),(b)$ is due to Cauchy's inequality, and $(c)$ follows from Lemma F.4.

The second inequality follows similar reasoning by applying Cauchy's inequality and Proposition D.1, Lemma F.2. $\quad\square$

# D. General MLE Analysis

In this section, we present a general proposition that characterize the performance of MLE.

MLE:

$$\hat{\theta}^{(k)} \in \left\{\theta: \sum_{(s_h,a_h,s_{h+1})\in\mathcal{D}}\log\mathcal{T}_{\theta,h}(s_{h+1}|s_h,a_h) \geq \max_{\theta'}\sum_{(s_h,a_h,s_{h+1})\in\mathcal{D}}\log\mathcal{T}_{\theta',h}(s_{h+1}|s_h,a_h)-\beta\right\}$$

The following proposition upper bounds the total variation distance between the conditional distributions over the future trajectory conditioned on the empirical history trajectories. This proposition is crucial to ensure that the model estimated by PSR-UCB is accurate on those sample trajectories.

**Proposition D.1.** *Let* $x_h = (s_h,a_h,s_{h+1})$. *Consider the following event*

$$\mathcal{E} = \left\{\forall k\in[K],\forall\theta\in\Theta^k,\ \sum_h\sum_{n<k}\mathrm{D_H^2}\left(\mathbb{P}_\theta^\mathrm{u}(x_h|s_{h-1}^{n,h},a_{h-1}^{n,h}),\mathbb{P}_{\theta^*}^\mathrm{u}(x_h|s_{h-1}^{n,h},a_{h-1}^{n,h})\right) \lesssim \log\frac{K|\Theta|}{\delta}\right\}.$$

*Then,* $\mathbb{P}(\mathcal{E}) \geq 1-\delta$.

*Proof.* Note that

$$D_H^2\left(\mathbb{P}_\theta^\pi(x_h|s_{h-1},a_{h-1}),\mathbb{P}_{\theta^*}^\pi(x_h|s_{h-1},a_{h-1})\right)$$

$$= 2\left(1 - \mathop{\mathbb{E}}_{x_h\sim\mathbb{P}_{\theta^*}^\pi(\cdot|s_{h-1},a_{h-1})}\sqrt{\frac{\mathbb{P}_\theta^\pi(x_h|s_{h-1},a_{h-1})}{\mathbb{P}_{\theta^*}^\pi(x_h|s_{h-1},a_{h-1})}}\right)$$

$$\overset{(b)}{\leq} -2\log\mathop{\mathbb{E}}_{x_h\sim\mathbb{P}_{\theta^*}^\pi(\cdot|s_{h-1},a_{h-1})}\sqrt{\frac{\mathbb{P}_\theta^\pi(x_h|s_{h-1},a_{h-1})}{\mathbb{P}_{\theta^*}^\pi(x_h|s_{h-1},a_{h-1})}},$$

where $(a)$ is due to Lemma F.2 and $(b)$ follows because $1 - x \leq -\log x$ for any $x > 0$.

Thus, the summation of the total variation distance between conditional distributions conditioned on $(\tau_h,\pi) \in \mathcal{D}_h^k$ can be upper bounded by

$$\sum_h\sum_{n<k}D_H^2\left(\mathbb{P}_\theta^u(x_h|s_{h-1}^{n,h},a_{h-1}^{n,h}),\mathbb{P}_{\theta^*}^u(x_h|s_{h-1}^{n,h},a_{h-1}^{n,h})\right) \leq -2\sum_h\sum_{n<k}\log\mathop{\mathbb{E}}_{x_h\sim\mathbb{P}_{\theta^*}^u(\cdot|s_{h-1}^{n,h},a_{h-1}^{n,h})}\sqrt{\frac{\mathbb{P}_\theta^u(x_h|s_{h-1}^{n,h},a_{h-1}^{n,h})}{\mathbb{P}_{\theta^*}^u(x_h|s_{h-1}^{n,h},a_{h-1}^{n,h})}},$$

In addition, we have

$$\mathop{\mathbb{E}}_{\forall n,h,\, x_{h+1}^{n,h}\sim\mathbb{P}_{\theta^*}^u(\cdot|s_{h-1}^{n,h},a_{h-1}^{n,h})}\left[\exp\left(\frac{1}{2}\sum_h\sum_{n<k}\log\frac{\mathbb{P}_\theta^u(x_h^{n,h}|s_{h-1}^{n,h},a_{h-1}^{n,h})}{\mathbb{P}_{\theta^*}^u(x_h^{n,h}|s_{h-1}^{n,h},a_{h-1}^{n,h})}\right.\right.$$

$$\left.\left.-\sum_h\sum_{n<k}\log\mathop{\mathbb{E}}_{x_h\sim\mathbb{P}_{\theta^*}^u(\cdot|s_{h-1}^{n,h},a_{h-1}^{n,h})}\sqrt{\frac{\mathbb{P}_\theta^u(x_h|s_{h-1}^{n,h},a_{h-1}^{n,h})}{\mathbb{P}_{\theta^*}^u(x_h|s_{h-1}^{n,h},a_{h-1}^{n,h})}}\right)\right]$$

$$= \frac{\mathop{\mathbb{E}}_{\substack{\forall n,h,\\ x_{h+1}\sim\mathbb{P}_{\theta^*}^u(\cdot|s_{h-1}^{n,h},a_{h-1}^{n,h})}}\left[\prod_h\prod_{n<k}\left(\frac{\mathbb{P}_\theta^u(x_h|s_{h-1}^{n,h},a_{h-1}^{n,h})}{\mathbb{P}_{\theta^*}^u(x_h|s_{h-1}^{n,h},a_{h-1}^{n,h})}\right)^{1/2}\right]}{\prod_h\prod_{n<k}\mathbb{E}_{x_h\sim\mathbb{P}_{\theta^*}^u(\cdot|s_{h-1}^{n,h},a_{h-1}^{n,h})}\left[\left(\frac{\mathbb{P}_\theta^u(x_h|s_{h-1}^{n,h},a_{h-1}^{n,h})}{\mathbb{P}_{\theta^*}^u(x_h|s_{h-1}^{n,h},a_{h-1}^{n,h})}\right)^{1/2}\right]}$$

$$= 1,$$

where the last equality is due to the conditional independence of $x_h$ given $(s_{h-1},a_{h-1})$.

Therefore, by the Chernoff bound, with probability $1-\delta$, we have

$$-2\sum_h\sum_{n<k}\log\mathop{\mathbb{E}}_{x_h\sim\mathbb{P}_{\theta^*}^u(\cdot|s_{h-1}^{n,h},a_{h-1}^{n,h})}\sqrt{\frac{\mathbb{P}_\theta^u(x_h|s_{h-1}^{n,h},a_{h-1}^{n,h})}{\mathbb{P}_{\theta^*}^u(x_h|s_{h-1}^{n,h},a_{h-1}^{n,h})}} \leq \sum_h\sum_{n<k}\log\frac{\mathbb{P}_{\theta^*}^u(x_h^{n,h}|s_{h-1}^{n,h},a_{h-1}^{n,h})}{\mathbb{P}_\theta^u(x_h^{n,h}|s_{h-1}^{n,h},a_{h-1}^{n,h})} + 2\log\frac{1}{\delta}.$$

Taking the union bound over $\Theta$, $k \in [K]$, and rescaling $\delta$, we have, with probability at least $1 - \delta$, $\forall k \in [K]$, the following inequality holds:

$$\sum_h\sum_{n<k}D_H^2\left(\mathbb{P}_\theta^u(x_h|s_{h-1}^{n,h},a_{h-1}^{n,h}),\mathbb{P}_{\theta^*}^u(x_h|s_{h-1}^{n,h},a_{h-1}^{n,h})\right)$$

$$\leq \sum_h\sum_{n<k}\log\frac{\mathbb{P}_{\theta^*}^u(x_h^{n,h}|s_{h-1}^{n,h},a_{h-1}^{n,h})}{\mathbb{P}_\theta^u(x_h^{n,h}|s_{h-1}^{n,h},a_{h-1}^{n,h})} + 2\log\frac{K|\Theta|}{\delta}$$

$$= \sum_h\sum_{n<k}\log\frac{\mathcal{T}_{\theta^*,h}(s_h^{n,h}|s_{h-1}^{n,h},a_{h-1}^{n,h})}{\mathcal{T}_{\theta,h}(s_h^{n,h}|s_{h-1}^{n,h},a_{h-1}^{n,h})} + \log\frac{\mathcal{T}_{\theta^*,h}(s_{h+1}^{n,h}|s_h^{n,h},a_h^{n,h})}{\mathcal{T}_{\theta,h}(s_{h+1}^{n,h}|s_h^{n,h},a_h^{n,h})} + 2\log\frac{K|\Theta|}{\delta}$$

$$\lesssim \log\frac{K|\Theta|}{\delta}$$

$\square$

# E. Existence of approximate low-rank factorization

**Theorem E.1.** *Let $\mathcal{X} \subset \mathbb{R}^{D_x}$ and $\mathcal{Y} \subset \mathbb{R}^D$ be compact sets. Let $\Theta$ be a finite model class. For each $(\theta, x) \in \Theta \times \mathcal{X}$, assume the conditional distribution $P_\theta(\cdot \mid x)$ admits a density $P_\theta(y \mid x)$ on $\mathcal{Y}$. We assume access to a sampling oracle that provides i.i.d. samples $y^{(1)}, \ldots, y^{(N)} \sim P_\theta(\cdot \mid x)$.*

**(A1) Boundedness.** *Assume there exists $M > 0$ such that $\sup_{\theta, x, y} P_\theta(y \mid x) \leq M$.*

**(A2) Smoothness and Boundary Vanishing.** *Assume there exists an integer $m > D$ and a constant $C_m$ such that $P_\theta(\cdot \mid x)$ is $m$-times differentiable on $\mathcal{Y}$. Crucially, assume that $P_\theta(\cdot \mid x)$ and its derivatives up to order $m - 1$ vanish on the boundary $\partial \mathcal{Y}$.*

*Then, for any radius $W \geq 1$, there exists a randomized algorithm that constructs feature maps $\hat{\phi}_\theta(x), \mu(y) \in \mathbb{R}^{2d}$ using $N$ samples, such that with probability at least $1 - \delta$, uniformly for all $(\theta, x, y) \in \Theta \times \mathcal{X} \times \mathcal{Y}_0$:*

$$\left| P_\theta(y \mid x) - \hat{\phi}_\theta(x)^\top \mu(y) \right| \leq C_{\text{trunc}} W^{D-m} + C_{\text{rand}} \text{Vol}(B_W) \left( \sqrt{\frac{1}{d}} + \sqrt{\frac{1}{N}} \right),$$

*where $C_{\text{trunc}}$ and $C_{\text{rand}}$ are constants depending on $m, D, \Theta, \mathcal{X}, \delta$ (explicit form in proof).*

*Proof.* We first construct the algorithm explicitly and then provide the error analysis to prove the bound.

**Construction of the Algorithm.** The algorithm proceeds as follows:

1. **Frequency Sampling:** Let $B_W = \{w \in \mathbb{R}^D : \|w\| \leq W\}$. Draw $d$ i.i.d. frequencies $w_1, \ldots, w_d$ uniformly from $B_W$. Let $\text{Vol}(B_W)$ denote the volume of this ball.

2. **Target Features ($\mu$):** Construct the feature map $\mu(y) \in \mathbb{R}^{2d}$ as:

$$\mu(y) = \frac{1}{\sqrt{d}} \left( \cos(2\pi w_1^\top y), -\sin(2\pi w_1^\top y), \ldots, \cos(2\pi w_d^\top y), -\sin(2\pi w_d^\top y) \right).$$

   *(Note the negative sign on the sine terms, corresponding to the inverse Fourier transform.)*

3. **Context Features ($\hat{\phi}$):** Draw $N$ samples $y^{(1)}, \ldots, y^{(N)} \sim P_\theta(\cdot \mid x)$. For each frequency $k \in \{1, \ldots, d\}$, compute the empirical Fourier coefficient:

$$\hat{g}_k = \frac{1}{N} \sum_{n=1}^{N} e^{-2\pi i w_k^\top y^{(n)}}.$$

Let $\hat{g}_k^{\text{r}} = \Re(\hat{g}_k)$ and $\hat{g}_k^{\text{i}} = \Im(\hat{g}_k)$. Construct $\hat{\phi}_\theta(x) \in \mathbb{R}^{2d}$ as:

$$\hat{\phi}_\theta(x) = \frac{\text{Vol}(B_W)}{\sqrt{d}} \left( \hat{g}_1^{\text{r}}, \hat{g}_1^{\text{i}}, \ldots, \hat{g}_d^{\text{r}}, \hat{g}_d^{\text{i}} \right).$$

**Analysis.** We decompose the error into three terms: Bias (Truncation), Variance (Random Features), and Estimation Error (Finite Samples).

$$|P_\theta - \hat{\phi}^\top \mu| \leq \underbrace{|P_\theta - \bar{P}_\theta|}_{(\text{I})} + \underbrace{|\bar{P}_\theta - \phi^\top \mu|}_{(\text{II})} + \underbrace{|\phi^\top \mu - \hat{\phi}^\top \mu|}_{(\text{III})}.$$

**Step 1: Truncation Error (I).** Let $g_\theta(w)$ be the Fourier transform of $P_\theta$ (extended by zero to $\mathbb{R}^D$). Due to Assumption (A2) (boundary vanishing), integration by parts yields the spectral decay $|g_\theta(w)| \leq c_{D,m} C_m \|w\|^{-m}$. The reconstruction $\bar{P}_\theta$ integrates frequencies only up to $W$. The tail mass is:

$$(\text{I}) \leq \int_{\|w\|>W} |g_\theta(w)| dw \leq \frac{S_{D-1} c_{D,m} C_m}{m - D} W^{D-m}.$$

**Step 2: Random Feature Error (II).** The term $\phi^\top \mu$ is a Monte Carlo approximation of the integral $\bar{P}_\theta = \int_{B_W} g_\theta(w)e^{2\pi i w^\top y}dw$. The variable $Z(w) = \text{Vol}(B_W)\Re(g_\theta(w)e^{2\pi i w^\top y})$ is bounded by $2\text{Vol}(B_W)$. Applying Hoeffding's inequality over $d$ features (and union bounding over the domain) gives:

$$(\text{II}) \leq \text{Vol}(B_W)\sqrt{\frac{2\log(4|\Theta||\mathcal{X}||\mathcal{Y}_0|/\delta)}{d}}.$$

**Step 3: Estimation Error (III).** This term arises from approximating $g_\theta(w_k)$ with $\hat{g}_k$ using $N$ samples.

$$(\text{III}) = \left|\sum_{k=1}^{d} \frac{\text{Vol}(B_W)}{d}(\hat{g}_k - g_k) \cdot (\dots)\right|.$$

Standard concentration (Hoeffding) on the $N$ samples shows that for any fixed $w_k$, $|\hat{g}_k - g_k| \lesssim \sqrt{1/N}$. Averaging these errors over $d$ frequencies does not reduce the variance below $O(1/\sqrt{N})$ because the samples are shared. The explicit bound (with union bounds) is:

$$(\text{III}) \leq \text{Vol}(B_W)\sqrt{\frac{2\log(4|\Theta||\mathcal{X}|d/\delta)}{N}}.$$

**Conclusion.** Summing the terms yields the stated bound. □

*Proof of Theorem 6.2.* We define the effective dimension $\tilde{d}$ to be

$$\tilde{d} = \max_{\theta\in\Theta,h} \dim\{\hat{\phi}_{\theta,h}(s_h,a_h), (s_h,a_h)\in\mathcal{S}\times\mathcal{A}\} < \infty,$$

where $\hat{\phi}$ is constructed through Algorithm 2. Note that while the dimension of constructed feature $\hat{\phi}$ can be infinitely large, the effective dimension can be finite due to compact set $\mathcal{S}$ and finite set $\mathcal{A}$.

Then, for all analysis in Appendix B and Appendix C, we incorporate approximation error $\zeta$, and every inequality will have an additional term $\zeta$ that bound the difference between the model and its low-rank approximation. In addition, $d$ will be replaced by $\tilde{d}$ as $d$ is constructed and could be infinitely large.

Thus, we can derive

$$V^* - V^{\tilde{\pi}} \leq \epsilon + \tilde{O}\left(H^2\tilde{d}|\mathcal{A}|\sqrt{\tilde{d}}\zeta\right)$$

□

# F. Auxillary Lemmas

We first establish one of the basic lemmas in RL literature: value difference lemma, which upper bound the difference between two values under different models, rewards, and policies.

**Lemma F.1** (Value-difference lemma). *Let $\theta, \theta' \in \Theta$, policies $\pi, \pi'$, and nonnegative reward functions $r, r'$. Assume the value functions are uniformly bounded:*

$$0 \leq V^\pi_{\theta,r}(s) \leq B_r, \qquad 0 \leq V^{\pi'}_{\theta',r'}(s) \leq B_{r'} \quad \text{for all } s.$$

*Then the value difference admits the following two upper bounds:*

$$V_{\theta,r}^{\pi} - V_{\theta',r'}^{\pi'} \leq V_{\theta,r-r'}^{\pi} + \sum_{h=1}^{H} \mathbb{E}_{\theta,\pi}\left[\sum_{a} Q_{\theta',r',h}^{\pi'}(s_h, a)\big(\pi(a|s_h) - \pi'(a|s_h)\big)\right]$$

$$+ B_{r'} \sum_{h=1}^{H} \mathbb{E}_{\theta,\pi}[\mathrm{D}_{\mathrm{TV}}(\mathcal{T}_{\theta,h}(\cdot|s_h, a_h), \mathcal{T}_{\theta',h}(\cdot|s_h, a_h))], \tag{18}$$

$$V_{\theta,r}^{\pi} - V_{\theta',r'}^{\pi'} \leq V_{\theta',r-r'}^{\pi'} + \sum_{h=1}^{H} \mathbb{E}_{\theta',\pi'}\left[\sum_{a} Q_{\theta,r,h}^{\pi}(s_h, a)\big(\pi(a|s_h) - \pi'(a|s_h)\big)\right]$$

$$+ B_{r} \sum_{h=1}^{H} \mathbb{E}_{\theta',\pi'}[\mathrm{D}_{\mathrm{TV}}(\mathcal{T}_{\theta,h}(\cdot|s_h, a_h), \mathcal{T}_{\theta',h}(\cdot|s_h, a_h))]. \tag{19}$$

*Proof.* We prove (18); (19) follows by swapping $(\theta, \pi, r)$ and $(\theta', \pi', r')$.

For any $h$ and state $s$, recall $V_{\theta,r,h}^{\pi}(s) = \sum_{a} \pi(a|s) Q_{\theta,r,h}^{\pi}(s, a)$. Thus

$$V_{\theta,r,1}^{\pi}(s_1) - V_{\theta',r',1}^{\pi'}(s_1) = \sum_{a} \pi(a|s_1) Q_{\theta,r,1}^{\pi}(s_1, a) - \sum_{a} \pi'(a|s_1) Q_{\theta',r',1}^{\pi'}(s_1, a)$$

$$= \sum_{a} \big(\pi(a|s_1) - \pi'(a|s_1)\big) Q_{\theta',r',1}^{\pi'}(s_1, a) + \sum_{a} \pi(a|s_1)\big(Q_{\theta,r,1}^{\pi} - Q_{\theta',r',1}^{\pi'}\big)(s_1, a).$$

Expand the $Q$-difference using Bellman equations:

$$Q_{\theta,r,1}^{\pi}(s_1, a) = r(s_1, a) + \mathbb{E}_{s' \sim \mathcal{T}_{\theta,1}(\cdot|s_1, a)}\big[V_{\theta,r,2}^{\pi}(s')\big],$$

and similarly for $(\theta', r', \pi')$. This yields

$$\sum_{a} \pi(a|s_1)\big(Q_{\theta,r,1}^{\pi} - Q_{\theta',r',1}^{\pi'}\big)(s_1, a)$$

$$= \sum_{a} \pi(a|s_1)\big(r(s_1, a) - r'(s_1, a)\big) + \sum_{a} \pi(a|s_1)\Big(\mathcal{T}_{\theta,1} V_{\theta,r,2}^{\pi} - \mathcal{T}_{\theta',1} V_{\theta',r',2}^{\pi'}\Big)(s_1, a)$$

$$= \sum_{a} \pi(a|s_1)\big(r(s_1, a) - r'(s_1, a)\big) + \sum_{a} \pi(a|s_1)\mathcal{T}_{\theta,1}\big(V_{\theta,r,2}^{\pi} - V_{\theta',r',2}^{\pi'}\big)(s_1, a)$$

$$+ \sum_{a} \pi(a|s_1)\big(\mathcal{T}_{\theta,1} - \mathcal{T}_{\theta',1}\big) V_{\theta',r',2}^{\pi'}(s_1, a).$$

For the last term, use the standard TV inequality: for any bounded $u$ with $0 \leq u \leq B_{r'}$,

$$|\mathbb{E}_P[u] - \mathbb{E}_Q[u]| \leq B_{r'} \, \mathrm{D}_{\mathrm{TV}}(P, Q).$$

Since rewards are nonnegative, $V_{\theta',r',2}^{\pi'} \in [0, B_{r'}]$, hence

$$\big(\mathcal{T}_{\theta,1} - \mathcal{T}_{\theta',1}\big) V_{\theta',r',2}^{\pi'}(s_1, a) \leq B_{r'} \, \mathrm{D}_{\mathrm{TV}}(\mathcal{T}_{\theta,1}(\cdot|s_1, a), \mathcal{T}_{\theta',1}(\cdot|s_1, a)).$$

Combining the above displays and taking expectation over $(s_1, a_1) \sim (\theta, \pi)$ gives

$$V_{\theta,r,1}^{\pi}(s_1) - V_{\theta',r',1}^{\pi'}(s_1) \leq \mathbb{E}_{\theta,\pi}[r(s_1, a_1) - r'(s_1, a_1)] + \mathbb{E}_{\theta,\pi}\left[V_{\theta,r,2}^{\pi}(s_2) - V_{\theta',r',2}^{\pi'}(s_2)\right]$$

$$+ \mathbb{E}_{\theta,\pi}\left[\sum_{a} Q_{\theta',r',1}^{\pi'}(s_1, a)\big(\pi(a|s_1) - \pi'(a|s_1)\big)\right]$$

$$+ B_{r'} \, \mathbb{E}_{\theta,\pi}[\mathrm{D}_{\mathrm{TV}}(\mathcal{T}_{\theta,1}(\cdot|s_1, a_1), \mathcal{T}_{\theta',1}(\cdot|s_1, a_1))].$$

Iterating this inequality from $h = 1$ to $H$ (telescoping the value terms) yields (18), noting that $V_{\theta,r,H+1}^{\pi} \equiv V_{\theta',r',H+1}^{\pi'} \equiv 0$. $\qquad \square$

The following lemma characterize the relationship between the total variation distance and the Hellinger-squared distance. Note that the result for probability measures has been proved in Lemma H.1 in Since we consider more general bounded measures, we provide the full proof for completeness.

**Lemma F.2.** *Given two bounded measures $P$ and $Q$ defined on the set $\mathcal{X}$. Let $|P| = \sum_{x \in \mathcal{X}} P(x)$ and $|Q| = \sum_{x \in \mathcal{X}} Q(x)$. We have*

$$\mathtt{D}_{\mathtt{TV}}^2(P, Q) \leq 4(|P| + |Q|)\mathtt{D}_{\mathtt{H}}^2(P, Q)$$

*In addition, if $P_{Y|X}, Q_{Y|X}$ are two conditional distributions over a random variable $Y$, and $P_{X,Y} = P_{Y|X}P$, $Q_{X,Y} = Q_{Y|X}Q$ are the joint distributions when $X$ follows the distributions $P$ and $Q$, respectively, we have*

$$\mathbb{E}_{X \sim P}\left[\mathtt{D}_{\mathtt{H}}^2(P_{Y|X}, Q_{Y|X})\right] \leq 8\mathtt{D}_{\mathtt{H}}^2(P_{X,Y}, Q_{X,Y}).$$

*Proof.* We first prove the first inequality. By the definition of total variation distance, we have

$$\mathtt{D}_{\mathtt{TV}}^2(P, Q) = \left(\sum_x |P(x) - Q(x)|\right)^2$$

$$= \left(\sum_x \left(\sqrt{P(x)} - \sqrt{Q(x)}\right)\left(\sqrt{P(x)} + \sqrt{Q(x)}\right)\right)^2$$

$$\overset{(a)}{\leq} \left(\sum_x \left(\sqrt{P(x)} - \sqrt{Q(x)}\right)^2\right)\left(2\sum_x (P(x) + Q(x))\right)$$

$$\leq 4(|P| + |Q|)\mathtt{D}_{\mathtt{H}}^2(P, Q),$$

where $(a)$ follows from the Cauchy's inequality and because $(a + b)^2 \leq 2a^2 + 2b^2$.

For the second inequality, we have,

$$\mathbb{E}_{X \sim P}\left[\mathtt{D}_{\mathtt{H}}^2(P_{Y|X}, Q_{Y|X})\right]$$

$$= \sum_x P(x)\left(\sum_y \left(\sqrt{P_{Y|X}(y)} - \sqrt{Q_{Y|X}(y)}\right)^2\right)$$

$$= \sum_{x,y} \left(\sqrt{P_{X,Y}(x,y)} - \sqrt{Q_{X,Y}(x,y)} + \sqrt{Q_{Y|X}(y)Q(x)} - \sqrt{Q_{Y|X}(y)P(x)}\right)^2$$

$$\leq 2\sum_{x,y} \left(\sqrt{P_{X,Y}(x,y)} - \sqrt{Q_{X,Y}(x,y)}\right)^2 + 2\sum_{x,y} Q_{Y|X}(y)\left(\sqrt{Q(x)} - \sqrt{P(x)}\right)^2$$

$$= 4\mathtt{D}_{\mathtt{H}}^2(P_{X,Y}, Q_{X,Y}) + 2(|P| + |Q| - 2\sum_x \sqrt{P(x)Q(x)})$$

$$\overset{(a)}{\leq} 4\mathtt{D}_{\mathtt{H}}^2(P_{X,Y}, Q_{X,Y}) + 2(|P| + |Q| - 2\sum_x \sum_y \sqrt{P_{Y|X}(y)P(x)Q_{Y|X}(y)Q(x)})$$

$$= 8\mathtt{D}_{\mathtt{H}}^2(P_{X,Y}, Q_{X,Y}),$$

where $(a)$ follows from the Cauchy's inequality that applies on $\sum_y \sqrt{P_{Y|X}(y)Q_{Y|X}(y)}$. $\qquad\square$

**Lemma F.3** (Elliptical potential lemma). *Let $(\mathcal{F}_i)_{i \geq 0}$ be a filtration. For $i \geq 1$, let $Y_i \in \mathbb{R}^d$ be an $\mathcal{F}_i$-measurable random vector such that its conditional law given the past is allowed to be adaptive, i.e.,*

$$Y_i \mid \mathcal{F}_{i-1} \sim P_i(\cdot \mid X_i),$$

*where $X_i$ and the kernel $P_i(\cdot \mid X_i)$ are $\mathcal{F}_{i-1}$-measurable (in particular, they may be arbitrary functions of the history). Fix $\lambda > 0$ and define the (predictable) Gram matrices*

$$A_1 := \lambda I_d, \qquad A_i := \lambda I_d + \sum_{j < i} Y_j Y_j^\top \quad (i \geq 2).$$

*Let $f_i := \|Y_i\|^2_{A_i^{-1}} = Y_i^\top A_i^{-1} Y_i$ and its clipped version*

$$\tilde{f}_i := \min\{1, f_i\}.$$

*Then the following hold:*

**(i) Conditional one-step bound.** *For every $i \geq 1$,*

$$\mathbb{E}\left[\tilde{f}_i \mid \mathcal{F}_{i-1}\right] \ \leq \ 2\,\mathbb{E}\left[\log\left(\frac{\det(A_{i+1})}{\det(A_i)}\right) \mid \mathcal{F}_{i-1}\right].$$

**(ii) Telescoping (expectation) bound.** *Consequently,*

$$\mathbb{E}\left[\sum_{i=1}^n \tilde{f}_i\right] \ \leq \ 2\,\mathbb{E}\left[\log\left(\frac{\det(A_{n+1})}{\det(\lambda I_d)}\right)\right].$$

**(iii) Deterministic upper bound under bounded norms.** *If additionally $\|Y_i\| \leq L$ almost surely for all $i$, then*

$$\sum_{i=1}^n \tilde{f}_i \ \leq \ 2\log\left(\frac{\det(A_{n+1})}{\det(\lambda I_d)}\right) \ \leq \ 2d\log\left(1 + \frac{nL^2}{\lambda d}\right),$$

*and hence also*

$$\mathbb{E}\left[\sum_{i=1}^n \tilde{f}_i\right] \ \leq \ 2d\log\left(1 + \frac{nL^2}{\lambda d}\right).$$

*Proof.* We use two standard facts.

**Step 1: Clipping versus log.** For all $x \geq 0$,

$$\min\{1, x\} \leq 2\log(1 + x).$$

Indeed, if $x \in [0, 1]$ then $\log(1 + x) \geq x/2$; if $x \geq 1$ then $2\log(1 + x) \geq 2\log 2 \geq 1$.

**Step 2: Matrix determinant lemma.** By $A_{i+1} = A_i + Y_i Y_i^\top$ and the matrix determinant lemma,

$$\det(A_{i+1}) = \det(A_i)\left(1 + Y_i^\top A_i^{-1} Y_i\right) = \det(A_i)\left(1 + f_i\right),$$

hence

$$\log\left(\frac{\det(A_{i+1})}{\det(A_i)}\right) = \log(1 + f_i).$$

**Proof of (i).** Condition on $\mathcal{F}_{i-1}$. The matrix $A_i$ is $\mathcal{F}_{i-1}$-measurable by construction, while $Y_i$ is drawn conditionally from an arbitrary kernel measurable w.r.t. $\mathcal{F}_{i-1}$. Applying Step 1 with $x = f_i$ and then Step 2 gives

$$\tilde{f}_i = \min\{1, f_i\} \leq 2\log(1 + f_i) = 2\log\left(\frac{\det(A_{i+1})}{\det(A_i)}\right).$$

Taking conditional expectations w.r.t. $\mathcal{F}_{i-1}$ yields (i).

**Proof of (ii).** Sum the inequality in (i) over $i = 1, \ldots, n$ and use telescoping:

$$\sum_{i=1}^n \log\left(\frac{\det(A_{i+1})}{\det(A_i)}\right) = \log\left(\frac{\det(A_{n+1})}{\det(A_1)}\right) = \log\left(\frac{\det(A_{n+1})}{\det(\lambda I_d)}\right).$$

Finally take total expectation.

**Proof of (iii).** The first inequality is Step 1 + Step 2 summed over $i$ (no expectation needed). For the last inequality, note that $A_{n+1} \preceq \lambda I_d + \sum_{i=1}^{n} L^2 I_d = (\lambda + nL^2)I_d$, so

$$\det(A_{n+1}) \leq (\lambda + nL^2)^d \quad \Rightarrow \quad \log\left(\frac{\det(A_{n+1})}{\det(\lambda I_d)}\right) \leq d\log\left(1 + \frac{nL^2}{\lambda}\right) \leq d\log\left(1 + \frac{nL^2}{\lambda d}\,d\right),$$

which implies the displayed $2d\log(1 + \frac{nL^2}{\lambda d})$ bound (a slightly sharper bound follows from AM–GM on eigenvalues). $\square$

**Lemma F.4.** *Let $\Phi$ be a finite class of feature maps $\phi : \mathcal{X} \to \mathbb{R}^d$ satisfying $\|\phi(x)\|_2 \leq 1$ for all $x \in \mathcal{X}$. Let $\{x_n\}_{n=1}^{K}$ be a sequence of random variables such that*

$$x_n \sim P_n(\cdot \mid \mathcal{F}_{n-1}),$$

*where $\{\mathcal{F}_n\}$ is a filtration. Define*

$$\Lambda_k = \lambda I + \sum_{n=1}^{k} \phi(x_n)\phi(x_n)^\top.$$

*Then with probability at least $1 - \delta$, for all $\phi \in \Phi$ and all $k \leq K$,*

$$\sum_{n=1}^{k} \mathbb{E}_{X_n \sim P_n}\left[\min\left\{1, \|\phi(X_n)\|_{\Lambda_k^{-1}}^2\right\}\right] \leq \sum_{n=1}^{k} \min\left\{1, \|\phi(x_n)\|_{\Lambda_k^{-1}}^2\right\} + O\left(\log\frac{K|\Phi|}{\delta} + d^2\log\frac{K}{\lambda}\right).$$

*Furthermore, the RHS can be simplied to $\tilde{O}\left(d^2\log(K|\Phi|/\delta)\right)$.*

*Proof.* Since $\Lambda_k \succeq \lambda I$, we have $0 \preceq \Lambda_k^{-1} \preceq \frac{1}{\lambda}I$. Let $\mathcal{M}$ be an $\epsilon$-net (under Frobenius norm) of the set

$$\left\{M \succeq 0 : \|M\|_{\mathrm{op}} \leq \tfrac{1}{\lambda}\right\}.$$

Standard covering arguments imply

$$\log|\mathcal{M}| = O\left(d^2\log\tfrac{1}{\lambda\epsilon}\right).$$

Fix $\phi \in \Phi$, $M \in \mathcal{M}$, and $k \leq K$. Define

$$Y_n = \min\left\{1, \|\phi(x_n)\|_M^2\right\}, \qquad \mu_n = \mathbb{E}_{X_n \sim P_n}[Y_n].$$

Then $0 \leq Y_n \leq 1$, and

$$\mathrm{Var}(Y_n) \leq \mu_n.$$

By Bernstein's inequality for conditionally independent bounded random variables, with probability at least $1 - \delta$,

$$\sum_{n=1}^{k}(\mu_n - Y_n) \leq \sqrt{2\log(1/\delta)\sum_{n=1}^{k}\mu_n} + \tfrac{2}{3}\log(1/\delta).$$

Rearranging yields

$$\sum_{n=1}^{k}\mu_n \leq \sum_{n=1}^{k}Y_n + O\left(\log\tfrac{1}{\delta}\right).$$

Applying a union bound over $\phi \in \Phi$, $M \in \mathcal{M}$, and $k \leq K$, we obtain that with probability at least $1 - \delta$,

$$\sum_{n=1}^{k}\mathbb{E}\left[\min\{1, \|\phi(X_n)\|_M^2\}\right] \leq \sum_{n=1}^{k}\min\{1, \|\phi(x_n)\|_M^2\} + O\left(\log\frac{K|\Phi||\mathcal{M}|}{\delta}\right).$$

By construction of $\mathcal{M}$, for each $k$ there exists $M \in \mathcal{M}$ such that

$$\|M - \Lambda_k^{-1}\|_F \leq \epsilon.$$

Using $\|\phi(x)\|_2 \le 1$, we have

$$\left| \|\phi(x)\|_M^2 - \|\phi(x)\|_{\Lambda_k^{-1}}^2 \right| \le \epsilon.$$

Since the function $a \mapsto \min\{1, a\}$ is 1-Lipschitz,

$$\left| \min\{1, \|\phi(x)\|_M^2\} - \min\{1, \|\phi(x)\|_{\Lambda_k^{-1}}^2\} \right| \le \epsilon.$$

Therefore,

$$\sum_{n=1}^k \mathbb{E}\left[ \min\{1, \|\phi(X_n)\|_{\Lambda_k^{-1}}^2\} \right] \le \sum_{n=1}^k \min\{1, \|\phi(x_n)\|_{\Lambda_k^{-1}}^2\} + O\left( \log \frac{K|\Phi||\mathcal{M}|}{\delta} + k\epsilon \right).$$

Choosing $\epsilon = 1/K$ ensures $k\epsilon \le 1$ and

$$\log |\mathcal{M}| = O\left( d^2 \log \frac{K}{\lambda} \right),$$

which completes the proof. $\qquad \square$

# G. Experiment Details

We provide more details and training results in this section.

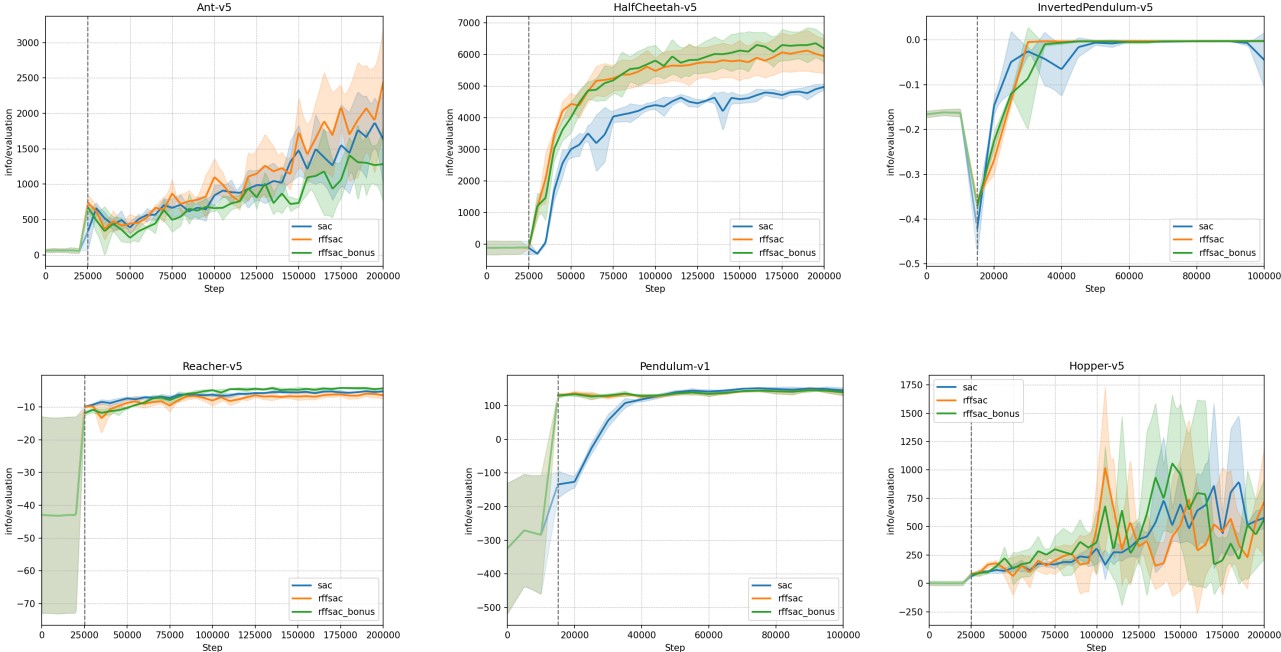

*Figure 1.* Evaluations over training. Steps before the dash line is uniform random exploration. y-axis is the value of one episode. Results are averaged across 4 different random seeds.

