# OpenReview forum: "Breaking the Computational Barrier: Provably Efficient Actor–Critic for Low-Rank MDPs"
_ICML.cc/2026/Conference — ICML 2026 regular_

### Official Review · Reviewer_9iwo · 2026-03-07

**Soundness:** 3
**Presentation:** 3
**Significance:** 2
**Originality:** 3
**Overall Recommendation:** 4
**Confidence:** 3

**Summary:**

This paper studies the computational and sample complexity of reinforcement learning in low-rank MDPs with unknown feature representations. The authors first establish a hierarchy among three commonly used RL oracles — policy evaluation (PE), policy planning (PP), and constrained planning (CP) — by measuring their complexity in terms of calls to a supervised learning (SL) oracle. Motivated by the computational advantage of PE, the authors propose Opt-AC, an optimistic actor–critic algorithm that relies solely on PE. The algorithm performs model learning via approximate MLE, constructs an optimistic critic using an elliptical exploration bonus defined from learned features, and updates the policy via mirror descent. The main theoretical result improves on the prior best (RAFFLE) by a factor of $d$. The paper also extends the framework to approximately low-rank MDPs, providing a conditional random Fourier feature (cRFF) construction that shows many practical transition models (e.g., Gaussian dynamics) admit such approximate low-rank structure. Experiments on standard Gym benchmarks show the method is competitive with SAC.

**Compliance With Llm Reviewing Policy:**

Affirmed.

**Final Justification:**

I maintain my positive score because the paper has some interesting results but future increasing my evaluation might require some more fundamental results.

**Key Questions For Authors:**

1. Can the authors provide wall-clock runtime comparisons between Opt-AC (with cRFF representation learning and bonus computation) and standard SAC?

2. Is there evidence that the SL-oracle complexity gap between PE and PP/CP is inherent, or could better reductions close the gap? The paper acknowledges this is open, but given the hierarchy is the key motivation, any additional insight would strengthen the narrative.

**Limitations:**

yes

**Strengths And Weaknesses:**

### Strengths

1.  The oracle hierarchy (PE vs. PP vs. CP) formalized via SL-oracle complexity is a clean and useful conceptual contribution, which provides a principled justification for designing algorithms around the PE oracle. This is a nice lens through which to view the computational landscape of low-rank MDP algorithms.

2. The main result achieves statistical efficiency while simultaneously requiring only PE, the weakest oracle in the hierarchy.

3. The algorithm avoids global planning subroutines entirely. The policy update via mirror descent admits a closed-form solution, and the critic construction is done through straightforward policy evaluation of the learned model with an exploration bonus. This is a structurally appealing design that aligns theory more closely with practice.

4. The cRFF construction (Theorem 6.1) and the resulting Theorem 6.2 are valuable for broadening the applicability of low-rank MDP theory. Showing that smooth transition kernels with vanishing boundary conditions admit approximate low-rank representations with polynomial sample complexity is a meaningful bridge between the theoretical framework and real-world environments.

5. The paper is generally well written. Table 1 is effective in summarizing the oracle hierarchy, computational properties, and sample complexities across methods. The distinction between oracle types and their regularity properties is clearly communicated.

### Weaknesses

1. The notation $\mathrm{SL}^m(\varepsilon)$ uses a superscript to denote the number of oracle calls, which can be misleading because it reads as an exponential quantity. A different notation (e.g., $\mathrm{SL}(m, \varepsilon)$ where $m$ is the call count) would be clearer.

2. The algorithm assumes access to a sampling oracle $\rho$ over the state-action space that provides sufficient coverage of the optimal policy's occupancy measure, with a coverage constant $C$ satisfying $\max \{d^{\star}_{\theta^\ast,h}(s), d^{\star}_{\hat{\theta}_k,h}(s)\} u(a) \leq C, \rho(s,a)$ for all $(s,a,h)$.

This assumption is not required in previous works. FLAMBE, Rep-UCB, and RAFFLE explore through their planning oracles (PP), which internally solve a global optimization problem to determine where to explore. These methods are "self-sufficient" for exploration and do not require any external coverage oracle. It would be beneficial to clarify that this is the price paid for using the weaker oracle (and ideally provide intuition on why this is required).

Critically, $C$ enters the final sample complexity bound. Yet Table 1 omits $C$ entirely. For example, how big is $C$ is in the worse case if $\rho$ is taken as the uniform distribution over state and action? I think this should be made more clear.

4. The sample complexity is stated as depending on $\log|\Theta|$. However, the paper defines $\Theta := \Phi \times \Psi$ (Section 3.1), so $\log|\Theta| = \log|\Phi| + \log|\Psi|$, meaning the bound depends on the complexity of *both* the state-action feature class $\Phi$ and the next-state feature class $\Psi$. This is natural for model-based approaches and is shared with FLAMBE, Rep-UCB, and RAFFLE. This should be made more clear in the comparison in Table 1: model-free methods (SpanRL, MOFFLE) write $\log|\Phi|$ and depend only on the forward feature class. The authors should clarify this distinction explicitly.

5. The experiments, while welcome, have some limitations. (i) The comparison is limited to SAC and the authors' own variants, with no comparison to other low-rank MDP algorithms (even those that are implementable, such as methods from Ren et al. (2022) or Zhang et al. (2022a), both of which are cited). (ii) The results are noisy, particularly for Hopper, where the standard deviation ($\approx 538$--$740$) is comparable to or exceeds half the mean, making it hard to draw conclusions.

---

> ### Author Rebuttal · Authors · 2026-03-31
>
> We thank the reviewer for the careful reading, constructive questions, and encouraging comments. We especially appreciate the reviewer’s positive assessment that the algorithm design aligns theory more closely to the practie. Below we provide our detailed responses to the key questions and other concerns.
>
> **A1:** Thanks for the question. Please refer to the rebuttal to Reviewer vVnW for wall-clock time report.
>
> **A2:** This is a very good question. While it remains open, we believe there are strong structural reasons suggesting such a gap may be difficult to remove. In particular, PP involves the Bellman optimality operator $Q\_h^\star(s,a) = r(s,a) + \mathbb{E}[\max\_{a'} Q\_{h+1}^\star(s',a')]$.  Intuitively, to control the error of $Q\_h^\star$ at level $\varepsilon$, one needs sufficiently accurate estimates of $Q\_{h+1}^\star(s',a')$ uniformly over all actions $a'$, and this requirement propagates backward through the H-step recursion. This is exactly why PP requires a substantially more accurate SL subroutine than PE, with the error rate degrading by factors involving $|\mathcal A|^{-H}$.  CP is even more challenging: Appendix A shows that it induces a bilevel optimization problem, and existing bilevel optimization results indicate that such problems are NP-hard in general (as discussed in line 746-752 in the paper), with only stationary-point guarantees known for practical solvers. Therefore, while a formal hardness separation remains open, the current evidence strongly suggests that the computational gap is not merely an artifact of analysis, but reflects genuine structural differences between PE and planning-based oracles.
>
> **Response to the weaknesses:**
>
> **(1) sampling oracle $\rho$:** The reviewer is right that this point should be clarified more explicitly. Prior PP-based works may appear not to require such an assumption, but that is largely because it is hidden inside a much stronger planning oracle. Our paper chooses to expose the sampling interface explicitly, because we aim for a more realistic sample-then-optimize implementation model rather than assuming an oracle that globally computes $Q(s, a)$-type objects over potentially infinite state spaces. We will clarify this motivation more explicitly in the revision
>
> Regarding the size of $C$, we view it mainly as a factor affecting the computational complexity of implementing the oracle, rather than the sample complexity. Intuitively, when $C$ is large, the sampling distribution $\rho$ provides poorer coverage, so the supervised-learning oracle used to implement policy evaluation must be solved to higher accuracy. In fact, with a mild refinement of the analysis, if the PE oracle is required to achieve accuracy on the order of $O(1/(C\sqrt{K}))$, then the dependence on $C$ can be absorbed into the oracle/computational side and does not need to appear in the final sample-complexity bound. Thus, $C$ should be interpreted mainly as a coverage parameter governing how hard the PE subroutine is to solve under the chosen $\rho$.
> Moreover, in many natural settings $C$ scales polynomially with the action-space size and an appropriate notion of state-space size or coverage ratio.
>
> **(2) Comparisons:** We thank the reviewer for the comments. Comparison to other methods such as Ren et al. (2022) and Zhang et al. (2022a) is indeed valuable, but making the comparison fully fair is not straightforward. Those works were evaluated on older environment versions that are not fully aligned with the current MuJoCo-v5 setup, and reproducing them requires substantial additional implementation effort. In particular, Zhang et al. (2022a) relies on contrastive representation learning and is significantly more computationally expensive. To quantify this, we ran a short 5,000-step speed test and report the environment steps per second below (on a new RTX 3090).
>
> Table of running time: steps per second
>
> | alg | Ant | HalfCheetah | InvertedPendulum | Pendulum | Reacher | Hopper | Overall |
> | --- | --- | --- | --- | --- | --- | --- | --- |
> | SAC | 291 | 316 | 347 | 339 | 337 | 332 | 327 |
> | cRFFSAC | 185 | 204 | 215 | 216 | 218 | 207 | 207 |
> | cRFFSAC+bonus | 172 | 195 | 201 | 206 | 203 | 198 | 196 |
> | CTRLSAC | 60 | 62 | 62 | 63 | 63 | 63 | 62 |
>
> For the second point, we agree that some results are noisy. We believe this is partly due to the intrinsic variability of the environment rather than being unique to our method. Prior benchmarking work has also reported very large variance for SAC on Hopper; for example, Wang et al. report 726.4 $\pm$ 675.5, which is of a similar order to what we observe under 4 random seeds. We will clarify this in the revision.
>
> Wang et al., Benchmarking model-based reinforcement learning.
>
> **(3)** We thank the reviewer for other suggestions, and we will adopt them.
>
>
> Thank you again for your insightful comments. We hope our responses addressed your concerns and would greatly appreciate your kind consideration in increasing your score.

---

> > ### Author Rebuttal · Reviewer_9iwo · 2026-04-03
> >
> > I appreciate the author's response. Regarding the assumption on the sampling distribution, I fully agree that with this assumption, this gives the computational benefit, which I still think is a good contribution. Then this is suggesting a certain tradeoff, and maybe this is beyond the scope of the paper to show that this certain tradeoff is unavoidable, but it would be a very interesting result if it exists. Regarding A2 I also agree that it might be challenging and beyond the scope of the paper. That said, I will remain my positive score because the paper has some interesting results but future increasing my evaluation might require some more fundamental results.

---

> > > ### Author Response · Authors · 2026-04-06
> > >
> > > We thank the reviewer for the thoughtful follow-up and for maintaining a positive score. We are glad that the clarification helped make the intended computational tradeoff clearer. We will include the necessary discussion in the revision to make these points more explicit.

---

### Official Review · Reviewer_vVnW · 2026-03-07

**Soundness:** 3
**Presentation:** 2
**Significance:** 3
**Originality:** 2
**Overall Recommendation:** 4
**Confidence:** 4

**Summary:**

The paper studies reinforcement learning in low-rank MDPs and proposes an actor–critic method designed for this setting. The key idea is to analyze RL algorithms through a hierarchy of optimization oracles and reduce the required computation to supervised learning primitives. Based on this perspective, the authors develop an optimistic algorithm that alternates between transition model estimation, policy evaluation, and policy improvement, where policy evaluation relies on the linear parameterization. The paper provides guarantees on both sample complexity and computational complexity. Finally, experiments evaluate a nonlinear neural-network variant of the proposed approach on standard benchmarks.

**Compliance With Llm Reviewing Policy:**

Affirmed.

**Final Justification:**

The paper proposes interesting ideas on the role of different operators for estimation in low-rank MDPs. During the rebuttal, additional experiments were included that help better characterize the sample complexity gains of the proposed method. Overall, the paper offers valuable perspectives, and in my view it leans toward acceptance, although I would not strongly oppose a rejection.

**Key Questions For Authors:**

- What are the precise assumptions that guarantee realizability of the Q-function under the linear model used in policy evaluation? In particular, how do the low-rank transition structure and the reward function interact to ensure that $Q_h(s,a)=\phi_h(s,a)^\top w_h$ can represent the value functions induced by the Bellman recursion?

- What is the exact algorithm used in the experiments, and how does it relate to the theoretical algorithm analyzed in the paper? In particular, it would be helpful to clarify how the practical implementation differs from the model-based optimistic actor–critic framework assumed in the theory.

- How does the theoretical algorithm perform empirically in practice? It would be interesting to see experiments evaluating the algorithm closer to the theoretical setting, for example by analyzing its sample efficiency or computational behavior relative to the theoretical guarantees.

**Limitations:**

- The policy evaluation step relies on a linear parameterization of the value functions of the form $Q_h(s,a)=r_h(s,a)+\phi_h(s,a)^\top w_h$, where the same features $\phi_h$ appear in the transition factorization. While this representation can be justified under low-rank transitions, the precise assumptions that guarantee realizability of the value functions under this model (and how the reward and transition structure interact to ensure this) are not clearly stated in the paper and remain somewhat implicit.

- The proposed algorithm assumes an explicit policy representation over a finite action set, which naturally fits discrete action spaces. However, the experimental evaluation is conducted on continuous-control benchmarks, and it is not entirely clear how the theoretical algorithm extends to continuous or parametrized policy settings.

**Strengths And Weaknesses:**

**Strengths**

- The paper explores the intersection between low-rank structure in transition dynamics and value-function representations in reinforcement learning. Low-rank MDPs provide a meaningful abstraction that has received increasing attention in the literature, so it is interesting to see further work investigating how low-rank structure can be exploited algorithmically and in theoretical analysis.

- The proposed framework adopts a simplified setup that allows the authors to obtain clean theoretical guarantees. Within this setting, the analysis of both sample complexity and computational complexity appears solid, and the resulting bounds are competitive with existing results in the low-rank MDP literature.

**Weaknesses**

- The policy evaluation reduction implicitly relies on a linear parameterization of the Q-function of the form $Q_h(s,a)=\phi_h(s,a)^\top w_h$, where the features $\phi_h$ are the same ones used to factorize the transition kernel. While this representation can be derived from the low-rank transition model, the paper does not clearly explain this modeling step or the assumptions required for it to hold. In particular, realizability of the value functions under this parameterization effectively requires that the Bellman recursion remains representable in the span of the transition features, which constrains how rewards and transition dynamics interact. This assumption is not stated explicitly in the main text and has to be inferred from derivations in appendices, making the modeling setup underlying the complexity results somewhat opaque.

- Some other modeling choices are not clearly described in the main text beyond linear Q parametrization. For example, the approximate low-rank extension based on the cRFF construction is introduced to motivate the expressiveness of the model class, but this construction is largely theoretical and not directly used by the algorithm. In practice, the implementation relies on neural network features, and the connection between the theoretical approximation result and the algorithm used in experiments remains somewhat unclear.  As a result, it is difficult to determine whether the empirical results provide evidence for the theoretical claims of the paper.

- The empirical section does not directly validate the main theoretical claims of the work. Important aspects of the theoretical analysis, such as the exploration strategy used by the algorithm, are not clearly described or analyzed in the experiments. Moreover, the reported metrics focus only on final return on benchmark tasks, whereas the theoretical contributions concern sample complexity and computational efficiency. No measurements of sample efficiency, data usage, or computational cost are provided. Furthermore, the experimental evaluation compares the proposed approach mainly with standard deep RL algorithms such as SAC and variants of the proposed method. However, algorithms from the low-rank MDP literature discussed in the paper are not included in the empirical comparison, making it difficult to assess whether the proposed method provides practical improvements over prior work in this setting. It also would have been interesting to see the performance of the theoretical algorithm in the experiments

- The analysis assumes access to a sampling distribution $\rho$ that provides sufficient coverage of the occupancy measures of both the optimal policy and the learned policies. While such coverage assumptions are common in theoretical analyses, in the current presentation this requirement appears somewhat implicit. It would be helpful if the paper stated and discussed this assumption more explicitly, clarifying its role in the analysis and its implications for the applicability of the algorithm in practice.

- The proposed algorithm assumes an explicit policy representation over a finite action set, which restricts the theoretical guarantees to discrete action spaces. Extending the approach to parametric policies or continuous action spaces is not discussed in the theory, despite the empirical evaluation being conducted on continuous-control benchmarks.

---

> ### Author Rebuttal · Authors · 2026-03-31
>
> We thank the reviewer for the careful reading, constructive questions, and encouraging comments. We especially appreciate the reviewer’s positive assessment that the theoretical part is solid. Below we provide our detailed responses to the key questions and other concerns.
>
> **A1:** We would like to clarify that our analysis **does not** assume that the Q-function satisfies $Q_h^\pi(s,a)=\phi_h(s,a)^\top w_h$. Instead, the only assumption on the reward is *known and bounded*. Therefore, we do not impose any additional structural assumption on how rewards interact with transitions, and the Q-function class is broad and general.
>
> **Regarding the implementation of policy evaluation:** it is introduced as an oracle abstraction, so its implementation details are not required in the main paper. However, in Appendix A (proof of Proposition A.2, lines 593–598), we explicitly derive the *loss function* used for policy evaluation, making PE *concrete and implementable* in practice.
>
> **Remarks on linear structure:** Since the low-rank structure appears only in the transition, only the conditional value $\mathcal{T}V_h^\pi(s,a) = \phi_h(s,a)^\top w_h$ admits linear structure. This is proved in Proposition A.2. Thus, the correct representation is that the Bellman expectation term is linear in the feature, while the reward remains an arbitrary known bounded function added separately. Moreover, due to section 6, such setting in fact captures a rich class of environments, since **any** smooth, bonded transitions admit approximate low-rank factorization.
>
> **A2:** The exact experimental algorithm is described in Section 7.1. Concretely, we build on top of the lvrep-rl codebase, use the last hidden layer of the neural network that learns transition dynamics as a learned representation $\phi_h(s,a)$, fit the transition model by minimizing $\\|M\phi(s,a)-s'\\|_2^2$ (corresponding to Gaussian-model MLE), and then parameterize the critic by a two-layer MLP on top of $\phi(s,a)$. The optimism bonus is built from $\phi(s,a)$ as well. So the relationship to the theory is present in the paper: the implementation instantiates the main structural ideas of the theory *representation learning, PE-style critic construction*, and *optimism from learned uncertainty*. We will revise the paper to make this relationship more clear.
>
> **A3:**  We would like to clarify that Section 7 already positions the experiments as evaluating the optimistic actor-critic design, which is directly motivated by the theory. For a more explicit empirical view of sample efficiency, please refer to the sample-efficiency table in our rebuttal to Reviewer 6j68, which reports the number of interaction steps required to reach a fixed performance threshold. We also provide the wall-clock training time below.
>
> Wall-Clock Training Time per Seed (Hours, averaged across 4 seeds, device: Intel i7-12700K CPU, an NVIDIA RTX 3080 Ti GPU)
>
> | alg | Ant | HalfCheetah | InvertedPendulum | Pendulum | Reacher | Hopper |
> | --- | --- | --- | --- | --- | --- | --- |
> | SAC | 1.48 | 1.04 | 0.89 | 0.54 | 0.53 | 1.03 |
> | cRFFSAC | 2.06 | 2.06 | 1.03 | 0.90 | 1.14 | 1.42 |
> | cRFFSAC+bonus | 2.48 | 2.50 | 0.73 | 0.84 | 0.70 | 1.40 |
>
> In summary, these results show that, for a fixed performance threshold, our method often requires fewer interaction steps than SAC while maintaining acceptable wall-clock runtime. We believe this provides empirical support for the main message of the theory: the proposed design achieves favorable sample efficiency with practical computational cost.
>
> **Response to other concerns:** **(1) cRFF:** While the cRFF construction is to establish the expressiveness of the approximate low-rank model class, the practical algorithm, using neural network features, is motivated by the **same** structural idea. Specifically, as discussed in Section 7.1 (lines 418–425), the cRFF feature map can be viewed as applying a fixed nonlinear transformation followed by a linear layer. From this perspective, using the learned hidden representation of a neural network is a natural practical replacement.  **(2) Comparison with prior low-rank-MDP algorithms:** We note that prior theoretical works in this line rely on much stronger oracles and are not practical (Uehara et al., 2022 Tan et al., 2025). **(3) $\rho$:** The assumption on $\rho$ is stated in Proposition 3.7. Its role is to ensure that the sampling distribution covers the state-action regions relevant to Bellman backup and uncertainty control under the learned policies and the optimal policy. **(4) finite actions:** While our theoretical guarantee is  stated for finite action spaces, the experiments serve a different purpose: they are included to test the generalizability and practical usefulness of the core design principles beyond the exact theoretical setting
>
> Thank you again for your insightful comments. We hope our responses addressed your concerns and would greatly appreciate your kind consideration in increasing your score.

---

> > ### Author Rebuttal · Reviewer_vVnW · 2026-04-01
> >
> > I thank the authors for the clear explanations and for the additional experiments on sample complexity, which help better convey the impact of the paper. Overall, most of my concerns have been addressed in one way or another, and I will increase my score. I provide a few additional comments in response to the revision:
> > * The role of the Q-function is now clearer after a closer reading of Proposition 3. Part of the discussion currently in the appendix could beneficially be incorporated into the main text, providing a clearer explanation of the role of the value function in relation to the low-rank assumption.
> > * While existing low-rank baselines rely on assumptions that may be theoretically restrictive, they could still be implemented and included for empirical comparison. Even if the assumptions do not strictly hold, such methods may perform well in practice, and including them would help assess the empirical advantages of the proposed approach.

---

> > > ### Author Response · Authors · 2026-04-02
> > >
> > > We thank the reviewer for the positive follow-up and for increasing the score.
> > >
> > > For the first point, we thank for the suggestion and will add a remark clarifying the low-rank structure and the resulting Q function class.  For the second point, we agree that including prior low-rank baselines would further strengthen the empirical section. We will incorporate this suggestion in the revision where feasible.

---

### Official Review · Reviewer_6j68 · 2026-03-17

**Soundness:** 2
**Presentation:** 3
**Significance:** 2
**Originality:** 3
**Overall Recommendation:** 4
**Confidence:** 3

**Summary:**

The paper studies low-rank MDPs, i.e. MDPs whose state transition kernel admits a low-rank decomposition. It formalizes complexity of algorithms through access to optimization oracles, of which the paper considers four: the supervised learning oracle, the policy evaluation oracle, the policy planning oracle, and constrained planning oracle. The paper measures the computational efficiency of an algorithm through the number and accuracy of supervised learning oracles, and then gives the SL-oracle complexity of the policy evaluation, policy planning, and constrained planning oracles.

The paper then presents an algorithm for low-rank MDPs, called Optimistic Actor-Critic with Representation Learning. At each iteration the algorithm collects exploratory data using the current policy, then uses supervised learning, maximum likelihood estimation to be specific, to learn a state transition kernel, then constructs a critic with an optimistic exploration bonus, and then finally updates the policy using KL-regularized policy improvement. The paper then gives a result on the sample complexity of this algorithm, which is better than that of RAFFLE by a factor of $d$.

Then the paper considers MDPs which are only approximately low-rank, argues that this class of MDPs is very expressive, and gives an approximation algorithm called "Conditional Random Fourier Features", which learns a low-rank representation of an MDP. Then the paper specifies the sample-complexity of using the Optimistic Actor-Critic algorithm with those approximate low-rank features.

Finally, the paper evaluates the combined algorithm in 6 MuJoCo environments, with and without the exploration bonus, and compares it to soft-actor critic.

**Compliance With Llm Reviewing Policy:**

Affirmed.

**Final Justification:**

My questions regarding which kind of environments the theory and algorithm work well has been addressed adequately in the rebuttal.

While I acknowledge that there is precedent in using 6 MuJoCo environments with 4 seeds each, I still do not think that the empirical results are particularly convincing in terms of the practical viability of the proposed algorithm.

Still, I have updated my score to "4: Weak Accept.", since the theoretical contribution of the paper is quite interesting.

**Key Questions For Authors:**

1. In which environments would the proposed algorithm probably work well, and which ones would it not?
2. While the theory is all about sample complexity, the new algorithm's sample complexity is not evaluated in the experiments, and not compared to the sample efficiency of other algorithms. Does the proposed algorithm beat other algorithms in terms of sample complexity?

**Limitations:**

- It is not made clear in which MDPs the algorithms is likely to work or what it's limitations are.
- Also it is not made clear in which way the theoretical sample complexity of the algorithm shows up in practice, in particular whether it is better than the sample complexity of other algorithms.

**Strengths And Weaknesses:**

Strengths:
- The paper is well and clearly written, with consistent mathematical notation.
- Theory seems to be correct, and provides sample complexity bounds for the proposed algorithm.
- The paper doesn't just present results on low-rank MDPs, but also considers the problem of learning a low-rank representation of an MDP.

Weaknesses:
- The algorithms presented in the paper have very weak experimental validation. The paper only evaluates in 6 MuJoCo environments, with 4 random seeds each. The error bars in Table 2 are often times overlapping, so the results are not significant at all. Even if the reported means were significantly different, then in two out of 6 environments SAC is (tied for) best, and there is no clear sign in those numbers that the new algorithms improve over SAC. So no practitioner can conclude from those experiments whether the new algorithms is useful or not.
- The paper doesn't specify in what kind of MDPs the proposed algorithms is likely to work and which ones it isn't.
- While the theory is all about sample complexity, the new algorithm's sample complexity is not evaluated in the experiments, and not compared to the sample efficiency of other algorithms.

Minor issues:
- 044: in RL the environment can be known, it might just be intractably large, as in chess or Go.
- 020: sounds a bit weird to say that real-world environments are becoming more complex, as they have always been very complex (self-driving cars, robotics, etc), and didn't really change?
- 033: "planing" should be "planning"
- 076: should be "we provide an affirmative answer to the above question..."
- 089: what is $\log | \Theta |$?
- 090: "numebr" should be "number"
- 099: would perhaps be good to already say here what $H, d, \mathcal{A}$ and $\tilde{O}$ are.
- 173: if one makes specific that $g$ needs to be measurable, then one should perhaps add that also $\mu_{\theta,h}$ needs to be measurable.
- 205: why is $f_w(x)$ in $\mathbb{R}$ and not in $\mathcal{Y}$?
- 175: even though one can guess what it is, it would perhaps also be good to define $Q^*_{\theta, r, h}$.
- 252: I think that it should be $d^*_{\theta, h}$.
- 261: should be "when the state space is compact".
- 287: maybe I missed it but what are the policies $\pi_0, ..., \pi_K$, and why would one return a uniform distribution over those?
- 326: step 8 in the algorithm is "end for", so shouldn't it make more sense to say "Steps 5-7"?
- 368: should be "the effective dimension".
- 386: aren't there 6 MuJoCo environments in the figure?
- 435: the paper only evaluates against a single baseline, so "against strong baselines" should be "against a strong baseline".

---

> ### Author Rebuttal · Authors · 2026-03-31
>
> We thank the reviewer for the careful reading, constructive questions, and encouraging comments. We especially appreciate the reviewer’s positive assessment that the paper is clearly written and also discussed the learning representation problem. Below we provide our detailed responses to the key questions and other concerns.
>
> **A1:** Thanks for the question. Our method is designed for environments whose transition dynamics are exactly or approximately low-rank, especially in continuous-state settings where the transition model is smooth and admits a compact feature representation. This is precisely the motivation of Section 6: Theorem 6.1 and Theorem 6.2 show that a broad class of smooth continuous-transition models admit approximate low-rank representations. The paper explicitly mentions Gaussian and near-Gaussian dynamics, smoothly parameterized physical systems, and locally linear stochastic transitions as examples of this regime.
>
> Accordingly, we expect the method to work well in environments with (1) finite state and action space, (2) smooth continuous transitions, and (3) low-dimensional latent transition structure.
>
> On the other hand, we do not expect the current theory to directly cover environments with deterministic transitions over continuous state space, or strongly misspecified low-rank structure. We will add a clearer discussion of this applicability regime and its limitations in the revision.
>
> **A2:**  Theoretically, our result proves that Opt-AC achieves $\widetilde O(H^5 d^3 |\mathcal A|^2 \log |\Theta|/\epsilon^2)$ sample complexity, improving the *best* prior low-rank result by a factor of $d$ (Table 1). This is the formal sense in which our method beats prior low-rank algorithms in sample complexity.
>
> Empirically, sample efficiency can be assessed by comparing the number of interaction steps required to reach the same target return. This is already reflected in the learning curves in Figure 1 of Appendix G. In particular, for a fixed performance threshold, our method often reaches that level in fewer steps than SAC. To make this comparison more explicit, we additionally report below the number of steps required to reach 90\% of the best achieved return in each environment (smaller is better).
>
> Table of Sample complexity: steps need to reach 90% of the best achieved value (indicated in the parenthesis, averaged across 4 seeds)
>
> | alg | Ant (3280) | HalfCheetah (6221) | InvertedPendulum (-0.06) | Pendulum (126.6) | Reacher (-5.43) | Hopper (1951) |
> | --- | --- | --- | --- | --- | --- | --- |
> | SAC | not reached | not reached | 26250 | 46250 | 136250 | 170000 |
> | cRFFSAC | 200000 | 131666 | 30000 | 16250 | 147500 | not reached |
> | cRFFSAC+bonus | not reached | 105000 | 31250 | 16250 | 93750 | 115000 |
>
> **Response to the weaknesses:** We would like clarify that the current experiments should not be interpreted as an empirical claim of superiority over SAC. Our empirical goal in this version is to demonstrate the **practical viability** of the two theory-motivated ingredients: *representation-based transition modeling and optimistic critic estimation* and the competitiveness against a strong baseline.
>
> The choice of 6 MuJoCo environments and 4 random seeds follows commonly used experimental settings in prior benchmark-oriented works such as Wang et al. (2019) and Zhang et al. (2022a), and was intended to provide a standard and comparable evaluation protocol rather than an exhaustive empirical study. We will revise the paper to make this positioning more explicit.
>
> Wang et al., Benchmarking model-based reinforcement learning.
> arXiv preprint arXiv:1907.02057, 39, 2019.
>
> Zhang et al., Making linear mdps practical via contrastive
> representation learning. In International Conference on
> Machine Learning, pp. 26447–26466. PMLR, 2022a.
>
> **Minors:** Thank you for catching these. We will fix the listed typos, adopt the suggestions, and provide answers to other questions: (1) 020: we will change it to "apply to more complex real-world environments"; (2) 089: $\Theta$ is the parameter space, and we assume finiteness WLOG, since otherwise we could use the covering number of $\Theta$; (3) 287: it should be $\pi^{(0)},...,\pi^{(K)}$ i.e., the algorithm returns a policy chosen uniformly from all policies generated during training. This averaging is theoretically standard and stabilizes the guarantee, since most iterates are provably good. Empirically, this behaves similarly to an exponential moving average and is often more stable than returning only the last policy.
>
> ---
>
> Thank you again for your insightful comments. We hope our responses addressed your concerns and would greatly appreciate your kind consideration in increasing your score.

---

> > ### Author Rebuttal · Reviewer_6j68 · 2026-04-02
> >
> > I thank the authors for their rebuttal.
> >
> > My questions regarding which kind of environments the theory and algorithm work well has been addressed adequately. However, while I acknowledge that there is precedent in using 6 MuJoCo environments with 4 seeds each, I still do not think that the empirical results are particularly convincing in terms of the practical viability of the proposed algorithm.
> >
> > Still, I will update my score to "4: Weak Accept.", since the theoretical contribution of the paper is quite interesting.

---

> > > ### Author Response · Authors · 2026-04-03
> > >
> > > We thank the reviewer's acknowledgment that the theoretical contribution is interesting. We also understand your remaining concerns regarding the empirical results and we will keep them in mind in the revision.
> > >
> > > Finally, we are grateful that you are willing to raise your score to 4 (Weak Accept). If possible, please kindly update your original score in the system as well.

---

### Official Review · Reviewer_sVJ2 · 2026-03-19

**Soundness:** 3
**Presentation:** 2
**Significance:** 2
**Originality:** 3
**Overall Recommendation:** 4
**Confidence:** 2

**Summary:**

A low-rank MDP is an MDP where the transition dynamics have a low rank structure. The authors first study the different optimization problems that low-rank MDP algorithms require oracles to solve---solving supervised learning problems, evaluating policies, performing planning, and solving planning problems over a constrained set of policies. The authors then show how each oracle can be written in terms of a supervised learning oracle in rank d MDPs. This allows for different low-rank MDP algorithms that rely on different oracles to be compared with each other.

Then, the authors introduce an actor-critic algorithm for solving low-rank MDPs that only requires one supervised learning oracle per iteration. They prove this algorithm achieves $\epsilon$ error in $\tilde{\mathcal{O}}(H^5d^3A^2\log(|\Theta|)/\epsilon^2)$ samples. They show that their algorithm is robust and achieves the same guarantees when the MDP is approximately low rank instead. They show that every MDP is approximately low rank by showing their Conditional Random Fourier Features algorithm outputs an low rank representation for any MDP satisfying a boundedness and smoothness condition. Finally, they empirically test their algorithm on MuJoCo.

**Compliance With Llm Reviewing Policy:**

Affirmed.

**Key Questions For Authors:**

1) Can you elaborate more on the comparison between your analysis and the Tan et al. (2025) algorithm that gets a $\tilde{\mathcal{O}}((dH^5 + dH^4 \log(|\Theta|))/\epsilon^2)$ sample complexity? Am I correct in understanding that they need $d^2 H |\mathcal{A}|^2$ times less samples, at the cost of calling $2^d$ times more supervised learning oracles? Is it correct for me to interpret this as the sample complexity is lower for them, but the run time is exponentially larger?
2) How long do you expect a supervised learning oracle to take when run? Since you only need to run the oracle on smooth and convex losses, would I be correct in saying it should be possible to converge to the optimal solution in polynomial time? Can you give a running time that isn't oracle dependent for Algorithm 1?
3) Are there natural settings in which we can expect the supervised learning oracle is possible to compute efficiently? For example, can we expect that calls to the supervised learning oracle are possible to evaluate in polynomial time when given the low-rank random Fourier features of the MDP outputted by Algorithm 2?
4) What is the definition of the term 'coverage constant' used in the statement of Proposition 3.7?
5) Since $\mathcal{T}$ is a probability distribution, why isn't the boundedness condition of Theorem 6.1 always satisfied with $M = 1$?

**Limitations:**

yes

**Strengths And Weaknesses:**

Strengths:
* The main ideas of the paper are simple. The main body is easy to read and looks correct, though I did not go through the math in the appendix in detail.
* This is not advertised as a main result of the paper, but I found Theorem 6.1 to be very interesting: that for all $\zeta > 0$, every MDP is $\zeta \sqrt{\log(|\mathcal{S}|)}$ close to a rank $1/\zeta^2$ MDP (It would be nicer if the dependence on parameters other than $\zeta$ was stated in the theorem statement in the main body directly instead of hidden in the middle of the proof in the appendix). Section 6 itself would be a useful contribution to the RL theory community, this is the first result that convinced me that low-rank MDPs are worth studying.
* The algorithm itself is very simple and looks easy to implement in practice.

Weaknesses:
* There isn't that big of a difference between SAC and the author's algorithms in the training runs shown in the appendix.
* Even if the number of oracle calls is small, there is no result that argues that each oracle call can be evaluated efficiently (for example in natural classes of MDPs).
* There is also still a big gap between the best known upper and lower bounds in the sample complexity of solving low-rank MDPs that prior work does better on (although it seems to require more compute, see question below).

Minor Errors:
* Line 33 should be "For instance, a constrained planning oracle requires" instead of "For instance, constrained planing oracle requires"
* Line 76 should be "We provide an affirmative answer" instead of "We provide affirmative answer"
* Line 328 should be "Step 4 (model learning) is a standard supervised learning problem" instead of "Step 4 (model learning) is standard supervised learning problems"
* In Table 1, $\Theta$ is swapped for $\Phi$ on the lines for MOFFLE and Opt-AC. The lower bound line has $A$ instead of $|\mathcal{A}|$.
* On line 568, a reference is undefined.

---

> ### Author Rebuttal · Authors · 2026-03-31
>
> We thank the reviewer for the careful reading, constructive questions, and encouraging comments. We especially appreciate the reviewer’s positive assessment that the algorithm is easy to implement, and Section 6 is interesting and valuable in its own right. Below we provide our detailed responses to the key questions and other concerns.
>
>
> **A1:** The high level idea is correct. But the details need more careful interpretation. Tan et al. 2025, obtain a better sample complexity bound, but under the strongest constrained planning (CP) oracle, whereas our algorithm uses only policy evaluation (PE). Under the best-known reductions in Proposition 3.7, PE requires only one supervised-learning call, while CP requires exponentially many in $d$ to *guarantee* the solution.
>
> We also note that their bound does *not* make the dependence on $\mathcal{A}$ explicit, which appears in the lower bound, so it is not straightforward to conclude that our sample complexity is simply $d^2 H |\mathcal{A}|^2$ worse by comparing the displayed terms alone. In fact, their result involves a term $\log|\mathcal{F}|$, where $\mathcal{F}$ is the Q-function class, which is non-trivially larger than the model class $\Theta$. Therefore, directly comparing our results term-by-term is not fully fair or informative.
>
> **A2:** For the PE reduction itself, the answer is essentially yes: once the feature is obtained, the PE problem becomes a standard convex smooth regression problem, so one expects polynomial-time optimization using standard methods. Likewise, if the MLE (model-estimation step) is a convex objective, or otherwise admits polynomial-time optimization, then the overall runtime of our algorithm is polynomial.
>
> **A3:** Yes. One concrete example is a linear Gaussian transition model of the form $\mathcal{T}(s' | s,a)=\mathcal{N}(M f(s,a), I)$, where $M$ is unknown and $f$ is a fixed embedding. Theorem 6.1 shows that such models admit an approximate low-rank representation via cRFF. In this setting, MLE reduces to an MSE objective over $M$, which is convex. Therefore, under such setting, both the model-learning step and the PE subroutine reduce to standard convex optimizations and are thus computable in polynomial time.
>
> **A4:** The definition is the first inequality in Proposition 3.7. Specifically, the coverage constant $C$ is the smallest constant such that the next-step state-action $(s',a')$ distribution induced from $\rho(s,a)$ after one transition with either policy $\pi$ or the uniform policy $u$ is dominated by $\rho(s',a')$. Intuitively, it measures how well the reference sampling distribution covers the state-action space under model transitions.
>
> **A5:** Because Theorem 6.1 is about the transition density value $\mathcal{T}\_{\theta,h}(s' \mid s,a)$, not necessarily total probability mass. A probability distribution integrates to one over $s'$, but its density at a point can exceed one in continuous spaces when the mass is concentrated, e.g. Gaussian distribution. Thus, the boundedness assumption is a uniform $L_\infty$-type bound on the density.
>
> **Response to the weaknesses:** We would like to clarify that this work is primarily a theoretical. The main contribution is a new statistical-computational tradeoff for low-rank MDPs: we improve the best known sample complexity for low-rank MDPs with unknown features, while relying only on policy evaluation (PE) rather than stronger planning-based primitives. The experiments are included mainly to demonstrate the **practical viability** of the two theory-motivated ingredients: *representation-based transition modeling* and *optimistic critic estimation*. We do not claim a universal empirical superiority over SAC. On the computational side, the reason we adopt SL-oracle complexity is that in rich function approximation, end-to-end runtime depends strongly on the particular learner and optimizer. We agree that the remaining upper-lower bound gap is still large, and we will state this more explicitly as an open problem.
>
> **Minors:** Thank you for catching these. We will fix the listed typos, specify $\Theta=\Phi\times\Psi$ in Table 1, and fix the undefined reference (Propositions A.2-A.4) at line 568.
>
> ---
>
> Thank you again for your insightful comments. We hope our responses addressed your concerns and would greatly appreciate your kind consideration in increasing your score.

---

> > ### Author Rebuttal · Reviewer_sVJ2 · 2026-04-04
> >
> > Thank you for your response, the rebuttal has addressed my concerns. I have read through the other reviews and have decided to maintain my score.

---

> > > ### Author Response · Authors · 2026-04-06
> > >
> > > Thanks for the positive support and your time throughout the review process. We are glad that our response addressed your concerns.

---

### Decision · Program_Chairs · 2026-04-30

**Decision:**

Accept (regular)

**Comment:**

The paper provides a solid theoretical advancement in the study of low-rank MDPs by formalizing a computational hierarchy of RL oracles. All reviewers agree that the theoretical framework is technically sound and that the reduction to supervised learning oracles provides a principled path toward more practical RL algorithms. While the empirical validation is somewhat limited, showing competitiveness rather than clear dominance over established baselines, the authors successfully addressed concerns regarding the necessity of their sampling assumptions as a fundamental trade-off for computational efficiency. The additional sample complexity and wall-clock data provided during the rebuttal helped characterize the practical impact of the work. Given the strong theoretical contribution and the interest from the RL community in making low-rank MDPs more tractable, the paper is recommended for acceptance.